# DISENTANGLING SOURCES OF RISK FOR DISTRIBUTIONAL MULTI-AGENT REINFORCEMENT LEARNING

## ABSTRACT

In cooperative multi-agent reinforcement learning, state transitions, rewards, and actions can all induce randomness in the observed long-term returns. This randomness is reflected from two risk sources: (a) *agent-wise* risk (i.e., how cooperative each teammate acts for a given agent) and (b) *environment-wise* risk (i.e., transition stochasticity). Although these two sources are both important factors for learning robust agent policies, prior works do not separate them or deal with only a single risk source which could lead to learning a suboptimal equilibrium only. In this paper, we propose **D**isentangled **RI**sk-sensitive **M**ulti-**A**gent reinforcement learning (DRIMA), a novel framework capable of disentangling risk sources. Our main idea is to separate risk levels in both centralized training and decentralized execution with a hierarchical quantile structure and quantile regression. Our experiments demonstrate that DRIMA significantly outperforms prior state-of-the-art methods across various scenarios in the StarCraft Multi-agent Challenge environment. Notably, DRIMA shows robust performance where prior methods learn only a suboptimal policy, regardless of reward shaping and exploration scheduling.

## 1 INTRODUCTION

In multi-agent reinforcement learning (MARL), centralized training with decentralized execution (CTDE) is one of the popular learning paradigms since it tackles the non-stationarity issues caused by the changing behaviors of the agents efficiently (Oliehoek et al., 2016). Under this paradigm, value-based CTDE has gained impressive attention. It consists of (i) training a joint action-value estimator with an access to global information in a centralized manner and (ii) decomposing the estimator into agent-wise utility functions which can be executed in a decentralized manner. Recently, several value-based CTDE methods have made training efficient and stable by designing the action-value estimator as flexible as possible while maintaining the execution constraint on decentralizability (Sunehag et al., 2018; Rashid et al., 2018; Son et al., 2019; Rashid et al., 2020a; Wang et al., 2020a).

However, despite the progress on value-based CTDE, agents still often fail to cooperate due to environment randomness, limited observation of agents, and time-varying policies of other agents, especially in highly stochastic environments. For such environments in single-agent settings, risk-sensitive reinforcement learning (RL) (Chow & Ghavamzadeh, 2014) has shown remarkable results by using policies that consider *risk* rather than simple expectations for return distribution caused by state transitions, rewards, and actions. Here, risk refers to the uncertainty over possible outcomes, and risk-sensitive policies act with a risk measure, such as variance or conditional value at risk (CVaR) (Chow & Ghavamzadeh, 2014). The main goal of this paper is to apply this risk-sensitive technique to value-based CTDE to learn more robust policies against various factors of uncertainty.

To this end, our motivating observation is that the randomness that must be dealt with in multi-agent settings come from two risk sources: (a) *agent-wise risk* and (b) *environment-wise risk*. Here, the agent-wise risk is about how other agents behave cooperatively for a given agent, and the environment-wise risk is about how the environment reacts randomly (i.e., stochasticity of the environmental dynamics) as illustrated in Figure 1. Unfortunately, existing value-based CTDE methods do not disentangle risk sources explicitly. For instance, recent approaches combining value-based CTDE with distributional RL (Qiu et al., 2021; Sun et al., 2021) learn the distribution of each agent-wise utility function but cannot separate different risk sources. This inability to disentangle risks can result to suboptimal policies under risk-sensitive RL; those methods cannot represent agent-wise

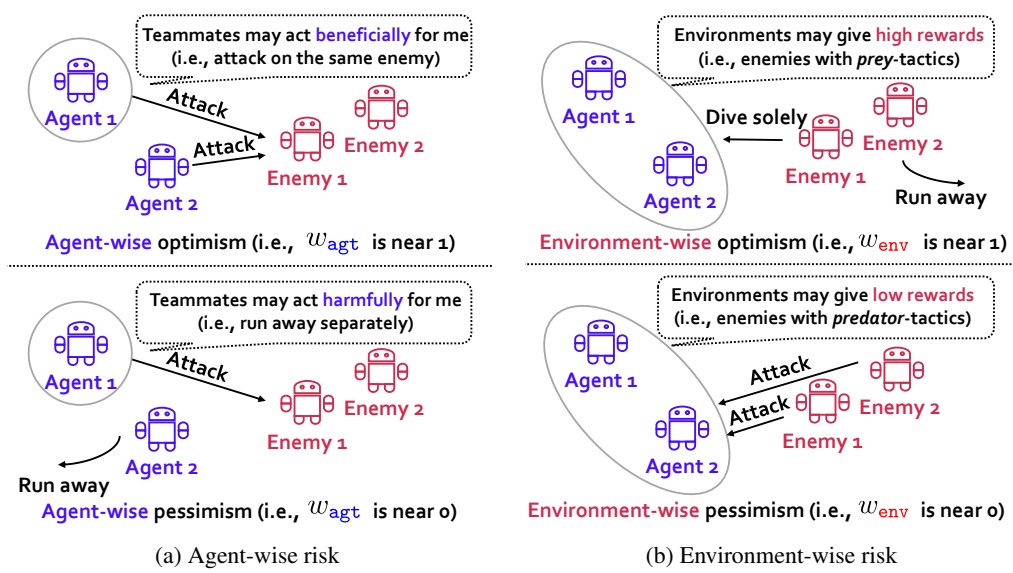

(a) Agent-wise risk        (b) Environment-wise risk

Figure 1: Two types of risk in MARL: (a) agent-wise risk and (b) environment-wise risk. Note that friendly agents learn a policy over time, while enemies in the environment act through a stationary distribution. Current value-based CTDE methods do not consider risks explicitly or tackle them in an entangled way which may lead to a suboptimal solution.

risk-seeking (cooperative) yet environment-wise risk-averse policies, which are likely to be of the favorable choices in most practical scenarios (i.e., surviving together as a team for a long time under the expectation that every teammate will cooperate for the sake of the team).

**Contribution.** In this paper, we present **D**isentangled **RI**sk-sensitive **M**ulti-**A**gent reinforcement learning (DRIMA), a novel framework on disentangling risk sources for distributional MARL. The main idea is to separate risk levels in both centralized training and decentralized execution with a hierarchical quantile structure and quantile regression. To be specific, unlike prior works which do not take into consideration different risk sources (Qiu et al., 2021; Sun et al., 2021), DRIMA considers agent-wise risk into each utility function and environment-wise risk into a joint true action-value estimator. Therefore, each utility function is encouraged to learn the action-value distribution with respect to other agents' cooperative policy, and the joint-action value estimator is trained to learn action-value distribution with

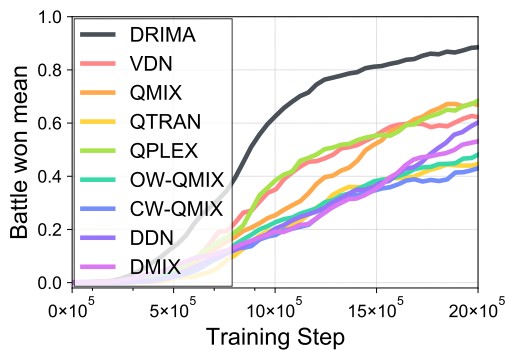

Figure 2: The median test win rate for DRIMA and eight value-based CTDE baselines, averaged across all 12 scenarios in our experiments. Results for each scenario are reported in Figure 4.

respect to the environment stochasticity. Thanks to the disentanglement of risk sources, DRIMA is able to obtain desirable policies more efficiently, while as aforementioned, it is difficult to set agent-wise risk-seeking yet environment-wise risk-averse policies in prior state-of-the-art methods.

We demonstrate the effectiveness of DRIMA on various scenarios in the StarCraft[1] Multi-Agent Challenge (SMAC) environment which is widely used as the environment for many MARL research (Samvelyan et al., 2019). As summarized in Figure 2, DRIMA significantly outperforms state-of-the-art methods, including distributional MARL methods (i.e., DDN and DMIX (Sun et al., 2021)) and non-distributional ones such as OW-QMIX, CW-QMIX, and QPLEX (Rashid et al., 2020a; Wang et al., 2020a). Notably, by disentangling risk sources, DRIMA shows impressive results regardless of reward shaping and exploration schedule, whereas existing works learn only a suboptimal policy (see Figure 4). We hope that our idea will motivate new future research directions such as safe MARL.

---

[1]StarCraft is a trademark of Blizzard Entertainment™.

## 2 PRELIMINARIES AND RELATED WORK

### 2.1 CENTRALIZED TRAINING WITH DECENTRALIZED EXECUTION

In this paper, we consider a decentralized partially observable Markov decision process (Oliehoek et al., 2016) represented by a tuple $\mathcal{G} = \langle \mathcal{S}, \mathcal{U}, P, r, O, N, \gamma \rangle$. To be specific, we let $s \in \mathcal{S}$ denote the true state of the environment. At each time step, an agent $i \in \mathcal{N} := \{1, ..., N\}$ selects an action $u_i \in \mathcal{U}$ as an element of the joint action vector $\boldsymbol{u} := [u_1, \cdots, u_N]$. It then goes through a stochastic transition dynamic described by the probability $P(s'|s, \boldsymbol{u})$. All agents share the same reward $r(s, \boldsymbol{u})$ discounted by a factor of $\gamma$. Each agent $i$ is associated with a partial observation $o \in \mathcal{O}$, according to some observation function $O(s, i) : \mathcal{S} \times \mathcal{N} \mapsto \mathcal{O}$, and an action-observation history $\tau_i$. Concatenation of the agent-wise action-observation histories is denoted as the overall action-observation history $\boldsymbol{\tau}$.

We consider value-based centralized training with decentralized execution (CTDE) where agents are trained in a centralized manner and executed in parallel without access to the global state $s$. Under value-based CTDE, we aim to train each agent-wise utility function $q_i(\tau_i, u_i)$ whose greedy actions are consistent with the greedy actions from the joint action-value estimator $Q_{\mathtt{jt}}(s, \boldsymbol{\tau}, \boldsymbol{u})$. Formally, the following *decentralization* condition should be satisfied:

$$\arg \max_{\boldsymbol{u}} Q_{\mathtt{jt}}(s, \boldsymbol{\tau}, \boldsymbol{u}) = \big[ \arg \max_{u_1} q_1(\tau_1, u_1), \ldots, \arg \max_{u_N} q_N(\tau_N, u_N) \big]. \quad (1)$$

To meet this condition, VDN (Sunehag et al., 2018) and QMIX (Rashid et al., 2018) impose structural constraint on the joint action-value function. To be specific, VDN expresses the joint action-value function as a linear summation of agent-wise utility functions. QMIX extends VDN by enforcing a monotonicity constraint on the joint action-value function and each agent-wise utility function. On the other hand, QTRAN (Son et al., 2019) uses soft regularization instead of structural constraints to satisfy the decentralization condition.

**Weighted QMIX.** Another prior work, namely Weighted QMIX (Rashid et al., 2020a), implicitly considers optimism in centralized training by weighting the loss of optimal actions, but it does not separate agent-wise and environment-wise risks. Rashid et al. (2020a) proposes Centrally-Weighted QMIX (CW-QMIX) and Optimistically-Weighted QMIX (OW-QMIX) which both have a strong theoretical capability of learning the largest class of true action-value functions up to date. They use a weighted projection that allows more emphasis to be placed on better joint actions. OW-QMIX applies weights to the losses according to the sign of the TD-error for each sample: [2]

$$\mathcal{L}_{\mathtt{OW-QMIX}}(s, \boldsymbol{\tau}, \boldsymbol{u}) = w(s, \boldsymbol{u})(Q_{\mathtt{tran}}(s, \boldsymbol{\tau}, \boldsymbol{u}) - y)^2,$$

$$y = r + \gamma Q_{\mathtt{jt}}^{\mathtt{target}}(s', \boldsymbol{\tau}', \boldsymbol{u}'_{\mathtt{opt}}), \quad w(s, \boldsymbol{u}) = \begin{cases} 1, & \text{if } Q_{\mathtt{tran}}(s, \boldsymbol{\tau}, \boldsymbol{u}) \leq y, \\ \alpha, & \text{otherwise,} \end{cases} \quad (2)$$

where $\alpha$ is a hyperparameter and $Q_{\mathtt{jt}}^{\mathtt{target}}$ is the fixed target network whose parameters are updated periodically from the original unconstrained true action-value estimator $Q_{\mathtt{jt}}$. They sample $(s, u, r, s')$, a tuple of experience transitions from a replay buffer and $\boldsymbol{u}'_{\mathtt{opt}}$ is the "optimal" action maximizing the utility functions $q_i(\tau'_i, u'_i)$ for $i \in \mathcal{N}$. The true action-value estimator $Q_{\mathtt{jt}}$ can accurately describe the true action-value using the unconstrained neural network. However, since this unconstrained network does not satisfy the decentralization condition (equation 1), we cannot extract the optimal action efficiently. On the other hand, the transformed action-value estimator $Q_{\mathtt{tran}}$ has limited expressive power because of its monotonic structure but can efficiently extract optimal actions. In order for $Q_{\mathtt{tran}}$ to accurately track the optimal action of $Q_{\mathtt{jt}}$, the weighting mechanism according to the sign of the TD-error assigns a higher weight for optimistic returns. Through the optimistic training, they prove that Weighted QMIX recovers the correct maximal action for any true-action value. We provide a more detailed description of other CTDE algorithms in Appendix C.

### 2.2 DISTRIBUTIONAL REINFORCEMENT LEARNING

Instead of training a scalar state-action estimator $Q^\pi(s, u)$, distributional RL represents an action-value as a random variable, denoted $Z^\pi(s, u)$. The distributional Bellman operator for policy

---

[2] Note that Rashid et al. (2020a) notates $Q_{\mathtt{tot}}$ instead of $Q_{\mathtt{tran}}$ in their paper.

evaluation in single-agent RL can then be expressed as follows:

$$Z^\pi(s,u) \stackrel{D}{=} R(s,u) + \gamma Z^\pi(S',U'), \tag{3}$$

where $S' \sim P(s'|s,u)$, $U' \sim \pi(u|s)$, and $A \stackrel{D}{=} B$ denotes the two random variables which follow the same probability distribution.

**IQN.** Implicit Quantile Networks (IQN) (Dabney et al., 2018a) is one of the widely-used single-agent distributional RL methods. IQN approximates the true action-value function using a quantile-based representation of the distributions. To be specific, each trained quantile represent a random variable $Z(s,u,\omega)$ which is a reparameterization of risk-level sample $\omega$ drawn from a uniform distribution $\mathcal{U}(0,1)$. To train the IQN, one can use the following Huber quantile regression loss (Huber, 1964):

$$\mathcal{L}_{\texttt{IQN}} = \begin{cases} |\omega - \mathbb{1}_{\delta>0}| \times \frac{1}{2}\delta^2, & \text{if } |\delta| \leq 1, \\ |\omega - \mathbb{1}_{\delta>0}| \times (|\delta| - \frac{1}{2}), & \text{otherwise}, \end{cases}$$

$$\delta = Z(s,u,\omega) - (r + \gamma Z^{\texttt{target}}(s', u'_{\texttt{opt}}, \omega')), \quad u'_{\texttt{opt}} = \arg\max_{u'} \mathbb{E}_w[Z^{\texttt{target}}(s, u', w)],$$

where $Z^{\texttt{target}}$ is the target network whose parameters are updated periodically from the original action-value estimator $Z$. $(s,u,r,s')$ is a tuple of experience transitions from a replay buffer. In distributional RL, weights are used to express level of rewards and next state transitions are generated from the randomness of the environment.

**Distributional MARL.** Recently, distributional RL has also been applied to the CTDE regime (Qiu et al., 2021; Sun et al., 2021), but they lack the disentanglement of risk sources. RMIX (Qiu et al., 2021) and DFAC (Sun et al., 2021) showed promising results by extending the agent-wise utility function from having deterministic variables to random variables. RMIX demonstrates the effectiveness of risk-sensitive MARL via distributional RL, but it is limited since it cannot represent policies which came from different risk sources (i.e., agent-wise risk-seeking yet environment-wise risk-averse policies). DFAC is another distributional MARL framework that uses *mean-shape decomposition* which separates the mean and variance parts of the utility functions to handle value function factorization. They propose DDN and DMIX as the DFAC variants of VDN and QMIX, respectively. DFAC extends IQN, a single-agent distributional RL algorithm, to learn the randomness of the environment. However, they do not consider the agent-wise risk that arises when learning networks with limited expressive power such as QMIX. Also, they only demonstrate risk-neutral policies, so it is questionable whether DFAC could handle risk-sensitive policies.

## 3 DRIMA: Disentangled risk-sensitive MARL

In this section, we propose a new method, called DRIMA, which aims at disentangling the two risk sources. In section 3.1, we discuss the connection between existing works which motivates DRIMA. In section 3.2, we propose the training objective of DRIMA for training our action-value estimators. Next, in section 3.3, we introduce a new architecture for the action-value estimators. The overall architectural sketch is given in Figure 3.

### 3.1 Motivation

Before presenting our framework, we re-visit existing risk-sensitive algorithms: distributional RL (Dabney et al., 2018a; Sun et al., 2021) and Optimistically-Weighted QMIX (OW-QMIX) (Rashid et al., 2020a) to discuss their connections and limitations, which are what motivate our method. In common, both methods utilize a weighting mechanism according to the sign of the TD-error. We understand that the entanglement comes from the TD-error that is used for computing weights, which play an important role in adjusting risk-sensitivity in both algorithms. To be specific, when the TD-error is positive, distributional RL and OW-QMIX interpret it as an optimistic sample in the perspective of environment-wise risk and agent-wise risk, respectively.

However, since reward and next state in the TD-target include both agent-wise and environment-wise randomness in an entangled way, agent-wise risk cannot be extracted separately enough from the sign of the TD-error. We observe that if environment-wise and agent-wise randomness co-exist, both distributional RL and OW-QMIX cannot learn the two randomness in a disentangled manner, which

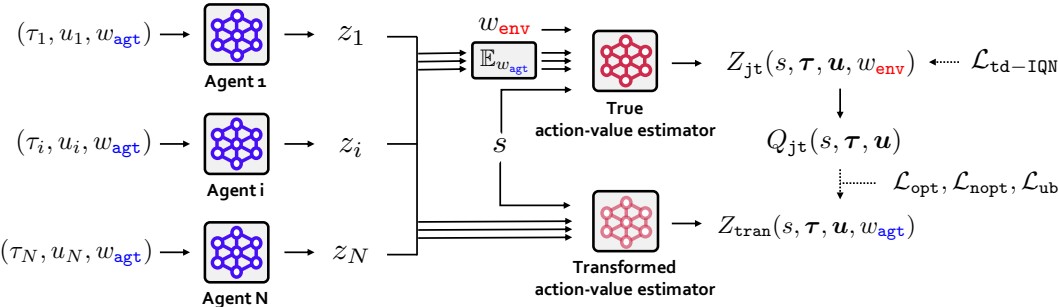

Figure 3: Architecture of DRIMA. Each agent network and true action-value estimator are built based on IQN and input a quantile of agent-wise risk $w_{\text{agt}}$ and environment-wise risk $w_{\text{env}}$, respectively. Along with the hierarchical quantile structure and quantile regression, the architecture is capable of disentangling risk sources explicitly. Further details for each component are provided in Appendix B.

may lead to a suboptimal policy (See section 4 for supporting experimental results); for instance, they cannot learn agent-wise risk-seeking yet environment-wise risk-averse policy. The fundamental similarity between distributional RL and optimistic training was also analyzed in fully decentralized MARL setting with independent learners (Rowland et al., 2021; Lyu & Amato, 2018). They show that Hysteretic Q-learning (Omidshafiei et al., 2017), which can be described as a fully distributed version of OW-QMIX, is related to distributional reinforcement learning.

Inspired by this, rather than handling risks in a entangled manner via a single weighting from a TD-error, we disentangle them via two separate action-value estimators with different roles; (i) true action-value estimator $Z_{\text{jt}}(w_{\text{env}})$ learns the joint action-value without structural constraints and captures environment-wise randomness with a risk level $w_{\text{env}}$, (ii) transformed action-value estimator $Z_{\text{tran}}(w_{\text{agt}})$ learns an action-value guided by the true-action value estimator with satisfying decentralization into the utility functions and captures agent-wise randomness with a risk level $w_{\text{agt}}$.

## 3.2 TRAINING OBJECTIVES

The training objective of DRIMA is twofold. First, DRIMA trains the true action-value estimator to approximate the distributional true action-value estimator $Z_{\text{jt}}$ based on the distributional Bellman operator (Equation 3). Next, the transformed action-value estimator attempts to imitate the behavior of the true action-value estimator through weights. Unlike the action-value estimator of DMIX, the true action-value estimator $Z_{\text{jt}}$ of DRIMA does not have any structural restrictions such as monotonicity. Therefore, $Z_{\text{jt}}$ can accurately represent only the randomness of the environment. On the other hand, the transformed action-value estimator $Z_{\text{tran}}$ limits its expressive power to satisfy the decentralization condition (equation 1). Like OW-QMIX, in order for $Z_{\text{tran}}$ to accurately track the optimal action of $Z_{\text{jt}}$, the weighting mechanism according to the sign of the TD-error assigns a higher weights for optimistic returns. However, we do not use the target $y$ as done in QMIX. Instead, we use $\mathbb{E}_{w_{\text{env}}}[Z_{\text{jt}}(w_{\text{env}})]$ to exclude the randomness of the environment in the target.

**Loss for the true action-value estimator.** The loss $\mathcal{L}_{\text{td-IQN}}$ trains the true action-value estimator $Z_{\text{jt}}$ with consideration of the environment-wise risk. To be specific, the loss is designed based on the IQN loss derived from Bellman operator (Equation 3) and the Huber quantile loss (Huber, 1964):

$$\mathcal{L}_{\text{td-IQN}} = \begin{cases} |w_{\text{env}} - \mathbb{1}_{\delta>0}| \times \frac{1}{2}\delta^2, & \text{if } |\delta| \leq 1, \\ |w_{\text{env}} - \mathbb{1}_{\delta>0}| \times (|\delta| - \frac{1}{2}), & \text{otherwise}, \end{cases}$$
$$\delta = Z_{\text{jt}}(s, \boldsymbol{\tau}, \boldsymbol{u}, w_{\text{env}}) - (r + \gamma Z_{\text{jt}}^{\text{target}}(s', \boldsymbol{\tau}', \boldsymbol{u}'_{\text{opt}}, w'_{\text{env}})),$$

where $Z_{\text{jt}}^{\text{target}}$ is the target network whose parameters are updated periodically from the original estimator $Z_{\text{jt}}$. Furthermore, $\boldsymbol{u}'_{\text{opt}}$ is the set of actions maximizing the utility functions $z_i(\tau'_i, u'_i)$ for $i \in \mathcal{N}$. For the $\mathcal{L}_{\text{td-IQN}}$, risk-level samples $w_{\text{env}}, w'_{\text{env}}$ are drawn from a uniform distribution $\mathcal{U}(0, 1)$.

**Loss for the transformed action-value estimator.** We propose $\mathcal{L}_{\text{opt}}, \mathcal{L}_{\text{nopt}}, \mathcal{L}_{\text{ub}}$ to encourage the transformed action-value estimator to track the value of the true action-value estimator while considering agent-wise risk $w_{\text{agt}}$. The formulation of each loss is given as follows, where we omit the

common function arguments $(s, \boldsymbol{\tau})$ for presentational simplicity:

$$\mathcal{L}_{\text{opt}} = \big(Z_{\text{tran}}(\boldsymbol{u}_{\text{opt}}, w_{\text{agt}}) - Q_{\text{jt}}(\boldsymbol{u}_{\text{opt}})\big)^2,$$

$$\mathcal{L}_{\text{nopt}} = \begin{cases} (Z_{\text{tran}}(\boldsymbol{u}, w_{\text{agt}}) - Q_{\text{jt}}(\boldsymbol{u}))^2, & \text{if } Z_{\text{tran}}(\boldsymbol{u}, w_{\text{agt}}) \le Q_{\text{jt}}(\boldsymbol{u}), \\ 2(1 - w_{\text{agt}}) \times (Z_{\text{tran}}(\boldsymbol{u}, w_{\text{agt}}) - Q_{\text{jt}}(\boldsymbol{u}))^2, & \text{otherwise}, \end{cases}$$

$$\mathcal{L}_{\text{ub}} = \begin{cases} (Z_{\text{tran}}(\boldsymbol{u}, w_{\text{agt}}) - Q_{\text{ub}}(\boldsymbol{u}))^2, & \text{if } Z_{\text{tran}}(\boldsymbol{u}, w_{\text{agt}}) \ge Q_{\text{ub}}(\boldsymbol{u}), \\ 0, & \text{otherwise}, \end{cases}$$

$$Q_{\text{jt}}(\boldsymbol{u}) = \mathbb{E}_{w_{\text{env}}}[Z_{\text{jt}}(\boldsymbol{u}, w_{\text{env}})], \quad Q_{\text{ub}}(\boldsymbol{u}) = \max(Q_{\text{jt}}(\boldsymbol{u}), Q_{\text{jt}}(\boldsymbol{u}_{\text{opt}})),$$

where $\boldsymbol{u}_{\text{opt}}$ is an "optimal" action maximizing the utility function $z(w_{\text{agt}})_i$ for $i \in \mathcal{N}$. In the subsequent paragraphs, detailed descriptions for each loss is provided.

Firstly, for optimal actions, $\mathcal{L}_{\text{opt}}$ encourages the transformed action-value estimator $Z_{\text{tran}}$ to follow the value of the true action-value estimator $Q_{\text{jt}}$. We can adjust the environment-wise risk sensitivity according to how we calculate the expected value $Q_{\text{jt}}$ from the distributional true action-value estimator $Z_{\text{jt}}$. If the expectation is calculated by sampling $w_{\text{env}}$ from a uniform distribution $\mathcal{U}(0, 1)$, the agents learn a risk-neutral policy for the environment-wise randomness. By changing this sampling distribution, we can learn the optimal policy for the desired environment-wise risk-sensitivity. For example, if we learn $Q_{\text{tran}}$ for $Q_{\text{jt}} = \mathbb{E}_{w_{\text{env}}}[Z_{\text{jt}}(w_{\text{env}})]$ calculated by sampling $w_{\text{env}}$ from a uniform distribution $\mathcal{U}(0, 0.25)$, agents take environment-wise risk-averse behaviors.

Secondly, for non-optimal actions, $\mathcal{L}_{\text{nopt}}$ aims to make $Z_{\text{tran}}$, which has a lower representational power, follow the true action-value estimator $Q_{\text{jt}}$ efficiently, utilizing agent-wise risk level as a weight. If the value of $Q_{\text{jt}}$ is greater than the value of the transformed action-value estimator $Z_{\text{tran}}$, then this is an optimistic sample where the corresponding action is likely to be the optimal action, so the transformed action-value estimator follows the true action-value estimator exactly. Conversely, for a relatively small true action-value, we softly ignore it and follow the true action-value estimator through a small weight whose value depends on the agent-wise risk level $w_{\text{agt}}$.

Finally, we add a loss function $\mathcal{L}_{\text{ub}}$ which makes an upper bound condition for $Z_{\text{tran}}$ for numerical stability because only the lower bound of $Z_{\text{tran}}$ exists in the $\mathcal{L}_{\text{nopt}}$ when $w_{\text{agt}} = 1$. By combining our loss functions $\mathcal{L}_{\text{opt}}, \mathcal{L}_{\text{nopt}}, \mathcal{L}_{\text{ub}}$ , we obtain the following objective which is minimized in an end-to-end manner to train the true and the transformed action-value estimators:

$$\mathcal{L} = \mathcal{L}_{\text{td-IQN}} + \lambda_{\text{opt}}\mathcal{L}_{\text{opt}} + \lambda_{\text{nopt}}\mathcal{L}_{\text{nopt}} + \lambda_{\text{ub}}\mathcal{L}_{\text{ub}}$$

where $\lambda_{\text{opt}}, \lambda_{\text{nopt}}, \lambda_{\text{ub}} > 0$ are hyperparameters controlling the importance of each loss function. The training algorithm of DRIMA is provided in Appendix A.

## 3.3 NETWORK ARCHITECTURES

Here, we introduce our architectures for the action-value estimators. First, we construct the estimators using the utility functions $z_1(w_{\text{agt}}), \ldots, z_N(w_{\text{agt}})$ with agent-wise risk level $w_{\text{agt}}$. Our main contribution in designing the estimators is twofold: (i) using IQN (Dabney et al., 2018a) for the true action-value estimator $Z_{\text{jt}}(w_{\text{env}})$ with environment-wise risk level $w_{\text{env}}$ and (ii) using a monotonic mixing network for the transformed action-value estimators $Z_{\text{tran}}$ with agent-wise risk level $w_{\text{agt}}$.

**Agent-wise utility function $z_i$.** In a partially observable setting, agents can learn better policies by using their action-observation history $\tau_i$ instead of their current observation. We represent each agent-wise utility function as a DRQN (Hausknecht & Stone, 2015) that receives action-observation history as input and outputs $z_i$ for each action and $w_{\text{agt}}$. The utility functions do not estimate the action-value, but aim to accurately extract the optimal action from the joint action-value.

**True action-value estimator $Z_{\text{jt}}$.** The estimator $Z_{\text{jt}}$ aims to express distributions with additional representation power; note that the transformed action-value estimator has limited power due to the decentralization condition (equation 1) (Rashid et al., 2018). For the true action-value estimator, we employ a feed-forward network that takes the state $s$ and set of utility functions $z_i$ for $i \in \mathcal{N}$. Since we use $z$ averaged over multiple $w_{\text{agt}}$ samples, the feed forward network is not conditioned on $w_{\text{agt}}$. To apply IQN for the true action-value estimator, we use an additional network $\phi$ that computes an embedding $\phi(w_{\text{env}})$ for the sample point $w_{\text{env}}$. We calculate the embedding of $w_{\text{env}}$ with cosine basis functions and utilize element-wise (Hadamard) product, as done in the IQN paper.

Table 1: Payoff matrix of the two-step game. $\mathcal{N}(\mu, \sigma^2)$ denotes the normal distribution with a mean of $\mu$ and a variance of $\sigma^2$. For the first step, agent-wise risk-seeking (cooperation) is crucial because the optimal action is $(A, A)$ but taking action $A$ is very risky because when the teammate chooses to be non-cooperative (i.e., by selecting action B or C), a catastrophic reward of -12 is provided. In the second step, handling environment-wise risks is highlighted since there are various combinations of mean and variance depending on joint actions, i.e., $\mathcal{N}(-1, 10)$ for the action $(C, C)$ and $\mathcal{N}(-1, 0)$ for the action $(B, B)$.

| $u_1$ \ $u_2$ | A | B | C |
|---|---|---|---|
| A | $\mathcal{N}(8, 0)$ | $\mathcal{N}(-12, 0)$ | $\mathcal{N}(-12, 0)$ |
| B | $\mathcal{N}(-12, 0)$ | $\mathcal{N}(0, 0)$ | $\mathcal{N}(0, 0)$ |
| C | $\mathcal{N}(-12, 0)$ | $\mathcal{N}(0, 0)$ | $\mathcal{N}(0, 0)$ |

(a) Payoff in the first step.

| $u_1$ \ $u_2$ | A | B | C |
|---|---|---|---|
| A | $\mathcal{N}(1, 5)$ | $\mathcal{N}(0, 5)$ | $\mathcal{N}(0, 5)$ |
| B | $\mathcal{N}(0, 5)$ | $\mathcal{N}(-1, 0)$ | $\mathcal{N}(-1, 5)$ |
| C | $\mathcal{N}(0, 5)$ | $\mathcal{N}(-1, 5)$ | $\mathcal{N}(-1, 10)$ |

(b) Payoff in the second step.

Table 2: Test rewards and trained policy in the stochastic two-step matrix game for DRIMA, DMIX, and OW-QMIX with varying risk-sensitivity across twelve random seeds.

| ALGORITHM | RISK SENSITIVITY | | TEST REWARD | | | $1^{\text{ST}}$ STEP | | $2^{\text{ND}}$ STEP | |
|---|---|---|---|---|---|---|---|---|---|
| | AGENT | ENV. | MIN | MEAN | MAX | $u_1$ | $u_2$ | $u_1$ | $u_2$ |
| DRIMA | NEUTRAL | AVERSE | -1.00 | -1.00 | -1.00 | B OR C | B OR C | B | B |
| | NEUTRAL | NEUTRAL | -1.28 | 0.89 | 4.13 | B OR C | B OR C | A | A |
| | NEUTRAL | SEEKING | -6.65 | 2.34 | 10.99 | B OR C | B OR C | C | C |
| | SEEKING | AVERSE | 7.00 | 7.00 | 7.00 | A | A | B | B |
| | SEEKING | NEUTRAL | 6.03 | 8.98 | 11.07 | A | A | A | A |
| | SEEKING | SEEKING | 2.78 | 5.50 | 13.01 | A | A | C | C |
| DMIX | AVERSE | | -1.00 | -1.00 | -1.00 | B OR C | B OR C | B | B |
| | NEUTRAL | | -2.66 | 0.89 | 2.94 | B OR C | B OR C | A | A |
| | SEEKING | | 1.89 | 6.96 | 12.81 | A | A | C | C |
| OW-QMIX | NEUTRAL | | -3.04 | 0.63 | 2.98 | B OR C | B OR C | A | A |
| | SEEKING | | 1.95 | 6.94 | 12.87 | A | A | C | C |

**Transformed action-value estimator** $Z_{\texttt{tran}}$. The architecture of the transformed action-value estimator is largely the same as the mixing network of Rashid et al. (2018). The transformed action-value estimator is expressed as follows:

$$Z_{\texttt{tran}}(s, \boldsymbol{\tau}, \boldsymbol{u}, w_{\texttt{agt}}) = f_{\texttt{mix}}\big(z_1(\tau_1, u_1, w_{\texttt{agt}}), \ldots, z_N(\tau_N, u_N, w_{\texttt{agt}}); \theta_{\texttt{tran}}(s, w_{\texttt{agt}})\big),$$

where $\theta_{\texttt{tran}}(s, w_{\texttt{agt}})$ is a non-negative parameter obtained from state-dependent hypernetwork (Ha et al., 2017). A more detailed discussion is available in Appendix D.

## 4 STOCHASTIC TWO-STEP MATRIX GAME

**Setup.** In this section, to showcase that DRIMA is indeed able to disentangle risks, we conduct experiments in a stochastic two-step matrix game which is a diagnostic illustrative environment widely used in the MARL community (Rashid et al., 2018; Son et al., 2019; Sun et al., 2021). In this game, two agents play a two-step matrix game where they select one action in $\{A, B, C\}$ and receive the shared global reward by a payoff matrix which should be maximized. As shown in Table 1, our matrix game is modified from the matrix game employed by Son et al. (2019) in order to compare the capability of different methods in handling risks. In particular, the first and the second steps are designed to assess whether an algorithm is able to handle agent-wise and environment-wise risks, respectively. Depending on the preference of a practitioner (i.e., environment-wise risk-seeking or risk-neutral), the proper policy must be learned. We compare DRIMA, DMIX (Sun et al., 2021) and OW-QMIX (Rashid et al., 2020a) with varying risk-sensitivity, conducted over 50k steps. We employ a full exploration scheme (*i.e.*, $\epsilon = 1$ in $\epsilon$-greedy) so that all available states will be visited.

**Analysis.** As shown in Table 2, we observe that DRIMA is able to represent diverse type of risk-sensitive policies. To be specific, agent-wise risk-seeking DRIMA selects action $A$ which requires strong cooperation of a teammate in the first step, while agent-wise risk-neutral DRIMA selects

$B$ or $C$ because it takes into consideration the noncooperation of a teammate. One can find the effectiveness of environment-wise risk-sensitivity via DRIMA from the variance of the test reward. Specifically, since there exist large variations in payoffs at the second step, environment-wise risk-seeking DRIMA decides to take risky actions like $(C, C)$ so that it results to high variance in the test reward. On the other hand, environment-wise risk-averse DRIMA shows low variance in the test reward because it takes conservative actions like $(B, B)$.

Whereas, DMIX and OW-QMIX have limited capability in adjusting risk-sensitivity. We observe that changing risk-sensitivity in DMIX and OW-QMIX affect agent-wise and environment-wise risks simultaneously. Namely, risk-seeking adjustment makes both agent-wise and environment-wise risk sensitivity become seeking. Therefore, DMIX and OW-QMIX may produce a suboptimal policy in environments which require different sensitivity for each risk source; note that agent-wise risk-seeking and environment-wise risk-neutral policies have the highest mean reward.

## 5 EXPERIMENTS

### 5.1 EXPERIMENTAL SETUP

**Environments.** We mainly evaluate our method on the Starcraft Multi-Agent Challenge (SMAC) environment (Samvelyan et al., 2019). In this environment, each agent is a unit participating in combat against enemy units controlled by handcrafted policies. In the environment, agents receive individual local observation containing distance, relative location, health, shield, and unit type of other allied and enemy units within their sight range. The SMAC environment additionally assumes that a global state is available during the training of the agents. To be specific, the global state contains information on all agents participating in the scenario. We select four representative scenarios which are diversified with respect to (i) desirable tactics and (ii) level of difficulty: 5m_vs_6m, 3s_vs_5z, 8m_vs_9m, and MMM2. For (i), in 5m_vs_6m and 8m_vs_9m, focusing fire is a plausible strategy; meanwhile, kiting is fruitful for 3s_vs_5z. For (ii), we select MMM2 since it has heterogeneous agents which makes it more difficult to handle them compared to when handling homogeneous ones. Appendix D contains additional experimental details.

**Increased exploration and rewarding mechanisms.** To further verify the effectiveness of disentangling risks, we additionally consider more difficult types of scenarios with differing exploration schedules and distinct mechanisms of reward functions. We refer to these three different types of scenarios as v1, v2 (increased exploration), and v3 (increased exploration and negative rewards), respectively. Details are as follows: **v1**: This is the same environment which is used in Rashid et al. (2018). **v2**: This is a modified version of v1 where the exploration annealing schedule is changed from "$50k$" to "$500k$" time steps following setups in Rashid et al. (2020a). **v3**: This has increased exploration as v2 and a different reward shaping; the reward is determined not only by the enemy units' health (positive) but also by damages dealt by our agents (negative). Interestingly, in increased exploration (v2 and v3), it is likely to obtain more non-cooperative behaviors from teammates. This represents the importance of considering agent-wise risk explicitly. In v3 where negative reward exists, if risks are not disentangled well, units are induced to have a "selfish" behavior (e.g., running away) in order to avoid being damaged.

**Evaluation.** We run 32 test episodes without an exploration factor for every $4 * 10^4$-th time step. The percentage of episodes where the agents defeat all enemy units, i.e., *test win rate*, is reported as the performance metric of the algorithms. We report the median performance with shaded 25-75% confidence intervals with four random seeds. For visual clarity, we smooth all the curves by moving average filter with a window size of 4.

### 5.2 RESULTS

**Comparisons to baselines.** We evaluate DRIMA compared to the eight baselines, including non-distributional (i.e., VDN, QMIX, QTRAN, QPLEX, OW-QMIX, and CW-QMIX) and distributional CTDE methods (i.e., DDN and DMIX). In Figure 2, we present the aggregated result which is the median test win rate averaged across all 12 scenarios (i.e., 4 Maps with 3 different versions) in our experiments. Along with Figure 4, which shows performance on each scenario, one can find that DRIMA achieves the state-of-the-art performance both sample-efficiently and asymptotically, while the second-best method varies for each scenario. For relatively easy tasks such as 5m_vs_6m,

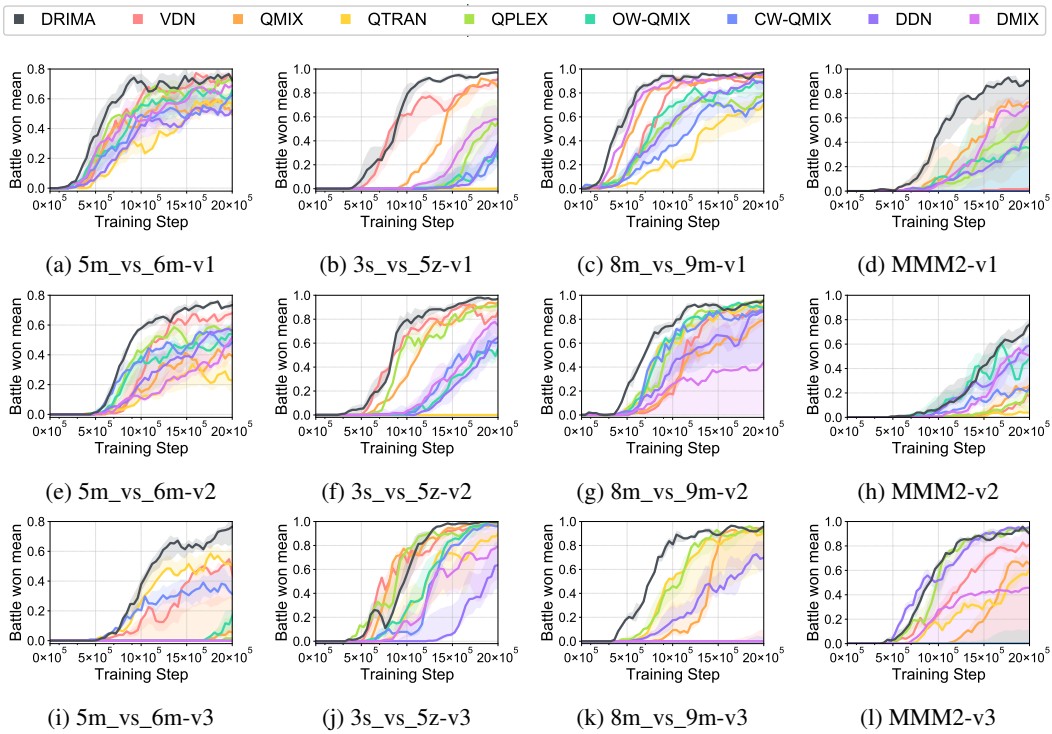

(a) 5m_vs_6m-v1  (b) 3s_vs_5z-v1  (c) 8m_vs_9m-v1  (d) MMM2-v1

(e) 5m_vs_6m-v2  (f) 3s_vs_5z-v2  (g) 8m_vs_9m-v2  (h) MMM2-v2

(i) 5m_vs_6m-v3  (j) 3s_vs_5z-v3  (k) 8m_vs_9m-v3  (l) MMM2-v3

Figure 4: Median test win rate with 25%-75% percentile over four random seeds, comparing DRIMA with eight baselines.

3s_vs_5z, and 8m_vs_9m, we set (agent-wise risk-sensitivity, environment-wise risk-sensitivity) to (seeking, neutral). For a hard task like MMM2, we set to (seeking, seeking). The rationale of this setting is that for easy tasks, considering environment-wise risks does not crucially affect the search for the optimal strategy, but for hard tasks which require extensive exploration, it is much beneficial to consider environmental risks; environment-wise risk-averse behavior is likely to avoid dangerous actions, which makes the agents less explorative.

Notably, as shown in Figure 4e-4l, DRIMA obtains significant gains in v2 and v3 , where separating agent-wise risk and environment-wise risk is critical; it is likely to reach a suboptimal equilibrium (i.e., selecting actions of running away) if risk sources are not disentangled in such environments. These results highlight the empirical effectiveness of our disentangling scheme.

Moreover, DRIMA shows consistent superiority regardless of difficulty and desirable tactics. Intriguingly, in relatively easier tasks (i.e., 5m_vs_6m), simple methods such as VDN show better results than more complex methods (i.e., DDN and DMIX[3]). We understand that a simple structure is more efficient for learning plausible strategies in such easy tasks. However, the reason why DRIMA is able to learn relatively easy tasks despite its sophisticated structure comes from that disentanglement of risk sources which makes useful learning signals for conquering easy tasks efficiently. We provide additional ablative studies and experiments on more various maps in Appendix F.

## 6  CONCLUSION

In this paper, we present DRIMA, a new distributional MARL framework that disentangles agent-wise risks and environment-wise risks in order to obtain optimal policies efficiently. Our main idea is to use disentangled risk sources along with the proposed hierarchical quantile structure and quantile regression. Through extensive experiments, we show that DRIMA achieves state-of-the-art performance thanks to the disentanglement of risk sources. We believe that DRIMA would guide a new interesting direction in the MARL community.

---

[3]In the original paper of DDN and DMIX (Sun et al., 2021), the results of 5m_vs_6m, 3s_vs_5z, and 8m_vs_9m were not reported.

## ETHICS STATEMENT

Cooperative MARL has potential real-world applications such as robot swarms (Huttenrauch et al., 2017), autonomous driving (Shalev-Shwartz et al., 2016), and sensor networks (Zhang & Lesser, 2011). However, there could be possible negative impacts if a malicious user designs a reward shaping that harms humans and society. For this reason, it is important to deeply consider the consequences of the agents' behaviors before releasing developed algorithms in the real world.

We have proposed a method that applies distributional reinforcement learning to CTDE algorithms. Our methods require additional research in terms of security. When a malicious external agent is added, there is a risk that our system is easily broken. Therefore, learning robust multi-agent systems against adversarial agents is an interesting future direction of research.

## REPRODUCIBILITY STATEMENT

One can find our reproducible code in the supplementary materials. All the used packages are along with the code. We remark that codebases of DRIMA and baselines including VDN, QMIX, QTRAN, OW-QMIX, CW-QMIX, DDN, and DMIX are based on the GitHub code of PyMARL[4]. As an evaluation benchmark, we employ StarCraft Multi-Agent Challenge (SMAC)[5] of which StarCraft version is SC2.4.6.2.69232. In our experiments, we use a single GPU (NVIDIA TITAN Xp) and 8 CPU cores (Intel Xeon E5-2630 v4).

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

# A DRIMA TRAINING ALGORITHM

The training algorithm for DRIMA is provided in Algorithm 1

---

**Algorithm 1** DRIMA algorithm

---

1: Initialize replay memory $\mathcal{B} \leftarrow \emptyset$ and target parameters $\theta^- \leftarrow \theta$
2: **for** episode = 1 to $M$ **do**
3:     Observe initial state $\boldsymbol{s}^0$ and observation $\boldsymbol{o}^0 = [O(\boldsymbol{s}^0, i)]_{i=1}^{N}$ for each agent $i$
4:     **for** $t = 1$ to $T$ **do**
5:         With probability $\epsilon$, each agent $i$ select an random action $u_i^t$
6:         Otherwise, set $u_i^t = \arg\max_{u_i^t} z_i(\tau_i^t, u_i^t, w_{\texttt{agt}})$ for each agent $i$ based on our agent-wise risk-sensitivity $w_{\texttt{agt}}$.
7:         Take action $\boldsymbol{u}^t$, and retrieve next state, observation and reward $(s^{t+1}, \boldsymbol{o}^{t+1}, r^t)$
8:         Store transition $(s^t, \boldsymbol{\tau}^{t+1}, \boldsymbol{u}^t, r^t, s^{t+1}, \boldsymbol{\tau}^{t+1})$ in $\mathcal{B}$
9:         Sample a transition $(s, \boldsymbol{\tau}, \boldsymbol{u}, r, s', \boldsymbol{\tau}')$ from $\mathcal{B}$
10:        Sample environment-wise risk levels $w_{\texttt{env}}, w'_{\texttt{env}}$ uniformly from $\mathcal{U}(0, 1)$
11:        Compute loss $\mathcal{L}_{\texttt{td-IQN}}$ for true action-value estimator from Section 3.2
12:        Sample environment-wise risk levels $w_{\texttt{env}}$ based on our environment-wise risk-sensitivity
13:        Sample agent-wise risk levels $w_{\texttt{agt}}$ uniformly
14:        Compute loss $\mathcal{L}_{\texttt{opt}}, \mathcal{L}_{\texttt{nopt}}$, and $\mathcal{L}_{\texttt{ub}}$ for transformed action-value estimator from Section 3.2
15:        Update $\theta$ by minimizing the losses
16:        Update target network parameters $\theta^- \leftarrow \theta$ with period $I$
17:     **end for**
18: **end for**

---

## B    DETAILED ARCHITECTURAL ILLUSTRATION

In this section, we provide detailed illustration for our architectural components: (i) agent-wise utility function, (ii) true action-value estimator, and (iii) transformed action-value estimator.

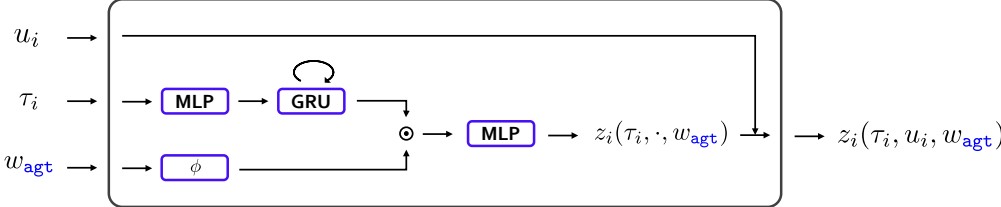

Figure 5: Agent-wise utility function. $\phi : [0, 1] \to \mathbb{R}^d$ denotes a quantile embedding function (Dabney et al., 2018a)

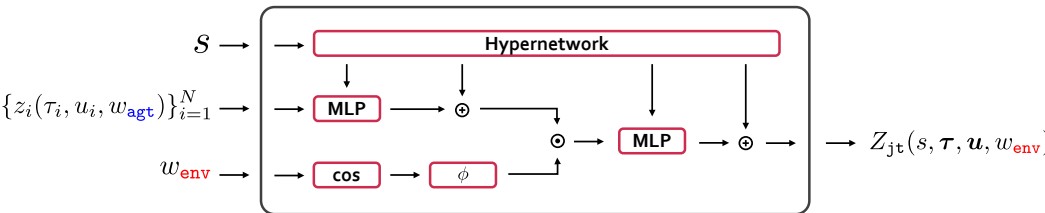

Figure 6: True action-value estimator. $\phi : [0, 1] \to \mathbb{R}^d$ denotes a quantile embedding function (Dabney et al., 2018a).

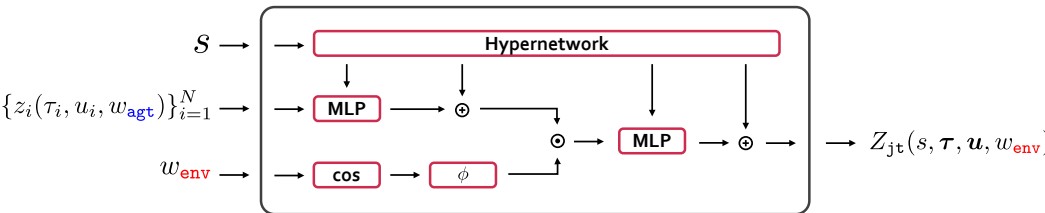

Figure 7: Transformed action-value estimator. $|\cdot|$ is employed to enforce monotonicity constraint (Rashid et al., 2018).

## C    RELATED WORK

### C.1    CENTRALIZED TRAINING WITH DECENTRALIZED EXECUTION

Centralized training with decentralized execution (CTDE) has emerged as a popular paradigm under the multi-agent reinforcement learning framework. It assumes the complete state information to be fully accessible during training, while individual policies allow decentralization during execution. To train agents under the CTDE paradigm, both policy-based (Foerster et al., 2018; Lowe et al., 2017; Du et al., 2019; Iqbal & Sha, 2019; Wang et al., 2020b; 2021) and value-based methods (Sunehag et al., 2018; Rashid et al., 2018; Son et al., 2019; Yang et al., 2020; Sun et al., 2021) have been proposed. At a high level, the policy-based methods rely on the actor-critic framework with independent actors to achieve decentralized execution. On the other hand, the value-based methods attempt to learn a joint action-value estimator, which can be cleverly decomposed into individual agent-wise utility functions.

Among the policy-based methods, COMA (Foerster et al., 2018) trains individual policies with a joint critic and solves the credit assignment problem by estimating a counterfactual baseline. MADDPG (Lowe et al., 2017) extends the DDPG (Lillicrap et al., 2015) algorithm to learn individual policies in a centralized manner on both cooperative and competitive games. MAAC (Iqbal & Sha, 2019) includes an attention mechanism in critics to improve scalability. LIIR (Du et al., 2019) introduces a meta-gradient algorithm to learn individual intrinsic rewards to solve the credit assignment problem. ROMA (Wang et al., 2020b) proposes a role-oriented framework to learn roles via deep RL with regularizers and role-conditioned policies. Finally, DOP (Wang et al., 2021) proposes factorized critic for multi-agent policy gradients.

Among the value-based methods, value-decomposition networks (VDN, Sunehag et al. 2018) learns a centralized yet factored joint action-value estimator by representing the joint action-value estimator as a sum of individual agent-wise utility functions. QMIX (Rashid et al., 2018) extends VDN by employing a *mixing network* to express a non-linear monotonic relationship among individual agent-wise utility functions in the joint action-value estimator. Qatten (Yang et al., 2020) introduces a multi-head attention mechanism for approximating the decomposition of the joint action-value estimator, which is based on theoretical findings. However, our method satisfies both theoretical guarantees and practical performance.

QTRAN (Son et al., 2019) has been proposed to eliminate the monotonic assumption on the joint action-value estimator in QMIX (Rashid et al., 2018). Instead of directly decomposing the joint action-value estimator into utility functions, QTRAN proposes a training objective that enforces the decentralization of the joint action-value estimator into the summation of individual utility functions. However, recent studies have found that despite its promise, QTRAN performs empirically worse than QMIX in complex MARL environments (Samvelyan et al., 2019; Rashid et al., 2020b).

One can find connections between DRIMA and QTRAN (Son et al., 2019), but DRIMA has a larger capacity to represent risk-sensitive policies. As for QTRAN, let's say $w_{\mathrm{agt}} = 1$ in DRIMA, then our loss functions $\mathcal{L}_{\mathrm{opt}}$ and $\mathcal{L}_{\mathrm{nopt}}$ become similar to QTRAN. However, DRIMA is capable of representing a wider range of risk-sensitive policies. For example, if the agent-wise risk level $w_{\mathrm{agt}}$ is 0.5, both $\mathcal{L}_{\mathrm{nopt}}$ and $\mathcal{L}_{\mathrm{opt}}$ follow the true action-value without weighting, which is agent-wise risk-neutral. This cannot be represented by QTRAN.

Recently, several other methods have been proposed to solve the limitations of QMIX. QPLEX (Wang et al., 2020a) takes a duplex dueling network architecture to factorize the joint value function. Unlike QTRAN, QPLEX learns using only one joint action-value network and a single loss function. Since they learn only a single estimator, it is impossible to learn agent-wise randomness and environment-wise randomness separately by applying distributional reinforcement learning as in our method. Also, Rashid et al. (2020a) proposed CW-QMIX and OW-QMIX, which use a weighted projection that allows more emphasis to be placed on better joint actions. Their methods apply weights to loss according to the sign of the TD-error for each sample, and it is theoretically guaranteed that the optimal policy can be learned for decentralizable tasks. However, whereas DRIMA learns the randomness of the environment using the weights from the TD-error, CW-QMIX and OW-QMIX only aim to consider the randomness of chosen actions. For this reason, they do not consider the distribution of the TD-error caused by the transition probability or the randomness of the reward in training, and it is not guaranteed that they can learn the optimal policy in stochastic environments.

On the other hand, the true action-value estimator of DRIMA learns only the randomness of the environment, and the transformed action-value estimator learns through the estimated true action-value, not the TD-target. Therefore, DRIMA completely separates the randomness of the environment and the randomness of the action in training.

## C.2 DISTRIBUTIONAL REINFORCEMENT LEARNING

Instead of training a scalar state-action estimator, distributional reinforcement learning focuses on representing a distribution over returns. Empirically, distributional reinforcement learning improves sample efficiency and performance.

A number of distributional reinforcement learning methods are proposed in a single-agent domain. Bellemare et al. (2017) proposed C51, which parameterizes the return distribution as a categorical distribution over a fixed set of equidistant points. In Dabney et al. (2018b), QR-DQN is proposed to estimate the distributions with a uniform mixture of N diracs and quantile regression. By estimating the quantile function with quantile regression, which has been shown to converge to the true quantile function, QR-DQN minimizes the Wasserstein distance to the distributional target. This estimation has been shown to converge to the true quantile function.

Implicit quantile Networks (IQN) (Dabney et al., 2018b) provides a way to learn an implicit representation of distributions by reparameterizing samples from base samples, typically $\omega \sim \mathcal{U}(0, 1)$. It learns a quantile function that maps from embeddings of sample probabilities to the corresponding quantiles, called implicit quantile networks. The quantiles are trained by Huber quantile regression loss (Huber, 1964).

Recently, distributional RL has also been applied to CTDE regime (Qiu et al., 2021; Sun et al., 2021), but they lack disentanglement of risk sources. RMIX (Qiu et al., 2021) and DFAC (Sun et al., 2021) showed promising results by extending agent-wise utility functions from deterministic variables to random variables. RMIX demonstrates the effectiveness of risk-sensitive MARL via distributional RL, but it is limited since they cannot represent policies, which have different risk levels relying on sources (i.e., agent-wise risk-seeking yet environment-wise risk-averse policies). DFAC is another distributional MARL framework with proposed mean-shape decomposition while employing IQN network for agent-wise utility function, but they only showcase risk-neutral policies. Da Silva et al. (2019) and Lyu & Amato (2018) have applied distributional learning in fully decentralized cases, not CTDE methods. Finally, Rowland et al. (2021) and Baker (2020) propose a method to learn cooperation using optimism in a fully distributed setting instead of CTDE.

## C.3 A FURTHER COMPARISON BETWEEN DRIMA AND OW-QMIX IN RISK-SENSITIVITY

In this section, we provide further backgrounds to understand why DRIMA is capable of disentangling agent-wise and environment-wise risks while OW-QMIX (Rashid et al., 2020a) cannot.

Before that, note that the risk-sensitivity is not mentioned explicitly in the original OW-QMIX paper, but we find that OW-QMIX can be viewed as an optimistic (agent-wise risk-seeking) algorithm due to its tight connection to hysterical Q-learning (Matignon et al., 2007; Omidshafiei et al., 2017), which study agent-wise optimism in the fully distributed MARL. In detail, one can find the connection from the loss function of hysterical Q-learning as follows:

$$
\begin{aligned}
\mathcal{L}_{\mathtt{hys-Q},i}(\tau, u) &= w_i(\tau, u)(q_i(\tau_i, u_i) - y_i)^2, \quad i \in \mathcal{N} \\
y_i &= r + \gamma q_i^{\mathtt{target}}(\tau_i', \arg\max_{u_i'} q_i(\tau_i', u_i')), \\
w(s, \boldsymbol{u}) &= \begin{cases} \alpha_{\mathtt{hys-Q}}, & \text{if } q_i(\tau_i, u_i) \leq y_i, \\ \beta_{\mathtt{hys-Q}}, & \text{otherwise}, \end{cases}
\end{aligned}
\tag{4}
$$

where the parameters $\alpha_{\mathtt{hys-Q}} > \beta_{\mathtt{hys-Q}} > 0$ are typically viewed as learning rate parameters, but here we equivalently view them as part of the loss. $q_i^{\mathtt{target}}$ is the fixed target network whose parameters are updated periodically from $q_i$. Unlike single-agent RL in a stationary environment, learned policies change through adjustment of the weights (agent-wise risk level) in MARL (Panait et al., 2006; 2008).

As with fully distributed settings, the idea of optimism is also applicable to value factorization methods such as QMIX (Rashid et al., 2018). QMIX is unable to represent joint action-value functions that are characterized as non-monotonic (Mahajan et al., 2019; Son et al., 2019), i.e., an agent's action-value ordering over its own actions depends on other agents' actions. To solve this problem, Rashid et al. (2020a) proposes Optimistically-Weighted QMIX (OW-QMIX) which assigns a higher weighting to those joint actions that are underestimated, and hence could be the true optimal actions in an optimistic (agent-wise risk-seeking) outlook, as follows:

$$
\begin{aligned}
\mathcal{L}_{\mathtt{OW-QMIX}}(s, \boldsymbol{\tau}, \boldsymbol{u}) &= w(s, \boldsymbol{u})(Q_{\mathtt{tran}}(s, \boldsymbol{\tau}, \boldsymbol{u}) - y)^2, \\
y &= r + \gamma Q_{\mathtt{jt}}^{\mathtt{target}}(s', \boldsymbol{\tau}', \boldsymbol{u}_{\mathtt{opt}}'), \\
w(s, \boldsymbol{u}) &= \begin{cases} 1, & \text{if } Q_{\mathtt{tran}}(s, \boldsymbol{\tau}, \boldsymbol{u}) \leq y, \\ \alpha, & \text{otherwise}, \end{cases}
\end{aligned}
\tag{5}
$$

where $\alpha < 1$ is a hyperparameter and $Q_{\mathtt{jt}}^{\mathtt{target}}$ is the fixed target network whose parameters are updated periodically from the original unconstrained true action-value estimator $Q_{\mathtt{jt}}$. They sample $(s, u, r, s')$, a tuple of experience transitions from a replay buffer and $\boldsymbol{u}_{\mathtt{opt}}'$ is the "optimal" action maximizing the utility functions $q_i(\tau_i', u_i')$ for $i \in \mathcal{N}$. One can find the loss function of hysterical Q-learning (Equation 4) and OW-QMIX (Equation 5) are similar, so that OW-QMIX can be understood as agent-wise optimistic algorithm like hysterical Q-learning.

To investigate similarities and differences between DRIMA and OW-QMIX to handle risk-sensitivity, we employ an one-step single-state matrix game. In this game, there is only one state, and all episodes finish after a single time-step. We find a connection between the DRIMA and OW-QMIX via following proposition:

**Proposition 1.** *Consider an deterministic-reward environment with a single state that terminated immediately. Then the loss $L_{\mathrm{nopt}}$ of DRIMA with parameters $w_{\mathrm{agt}}$ is equivalent to the Weighted QMIX with $\alpha = 2 * (1 - w_{\mathrm{agt}})$.*

*Proof.* In the simple one-step matrix game, the loss for Weighted QMIX reduces to

$$
\begin{aligned}
\mathcal{L}_{\mathtt{OW-QMIX}}(s, \boldsymbol{\tau}, \boldsymbol{u}) &= w(s, \boldsymbol{u})(Q_{\mathtt{tran}}(s, \boldsymbol{\tau}, \boldsymbol{u}) - r)^2, \\
w(s, \boldsymbol{u}) &= \begin{cases} 1, & \text{if } Q_{\mathtt{tran}}(s, \boldsymbol{\tau}, \boldsymbol{u}) \leq r, \\ \alpha, & \text{otherwise}, \end{cases}
\end{aligned}
\tag{6}
$$

since there is no next state to bootstrap from. In DRIMA, $Z_{\mathtt{jt}}$ estimates a deterministic value independent of $w_{\mathrm{env}}$ for the deterministic-reward environment. Then the loss $L_{\mathrm{nopt}}$ of DRIMA

reduces to

$$\mathcal{L}_{\text{nopt}}(s, \boldsymbol{\tau}, \boldsymbol{u}) = w(s, \boldsymbol{u})(Z_{\text{tran}}(s, \boldsymbol{\tau}, \boldsymbol{u}, w_{\text{agt}}) - Q_{\text{jt}}^{\text{DRIMA}}(s, \boldsymbol{\tau}, \boldsymbol{u}))^2,$$

$$Q_{\text{jt}}^{\text{DRIMA}}(s, \boldsymbol{\tau}, \boldsymbol{u}) = \mathbb{E}_{w_{\text{env}}}[Z_{\text{jt}}(s, \boldsymbol{\tau}, \boldsymbol{u}, w_{\text{env}})] \approx \mathbb{E}_{w_{\text{env}}}[r] = r,$$

$$w(s, \boldsymbol{u}) = \begin{cases} 1, & \text{if } Z_{\text{tran}}(s, \boldsymbol{\tau}, \boldsymbol{u}, w_{\text{agt}}) \leq Q_{\text{jt}}^{\text{DRIMA}}(s, \boldsymbol{\tau}, \boldsymbol{u}), \\ 2 * (1 - w_{\text{agt}}), & \text{otherwise}. \end{cases} \quad (7)$$

Now observe that if $\alpha = 2 * (1 - w_{\text{agt}})$ and $Z_{\text{tran}}$ is sufficiently trained, then two equations are equal. $\qquad\square$

This proposition says that DRIMA and OW-QMIX become equivalent in an deterministic-reward environment. However, if the environment is highly stochastic, then optimistic algorithms can induce misplaced optimism towards uncontrollable environment dynamics, leading to sub-optimal behavior. The optimistic MARL approaches ignore low returns, which are assumed to be caused by teammates' exploratory actions. This causes severe overestimation of action-values in stochastic domains (Wei & Luke, 2016; Rowland et al., 2021). Since reward and next state in the TD-target $y$ in Weighted QMIX include randomness of the environment, agent-wise risk cannot be extracted separately enough from the sign of the TD-error. Therefore, in DRIMA, we do not use the target $y$ as done in Weighted QMIX. Instead, we use $\mathbb{E}_{w_{\text{env}}}[Z_{\text{jt}}(w_{\text{env}})]$ to exclude the randomness of the environment in the target. The true action-value estimator $Z_{\text{jt}}$ of DRIMA does not have any structural restrictions such as monotonicity in value factorization methods. Therefore, $Z_{\text{jt}}$ can accurately represent only the randomness of the environment.

# D   EXPERIMENTAL DETAILS

The hyperparameters of training and testing configurations for VDN, QMIX, and QTRAN are the same as in the recent GitHub code of SMAC [6] (Samvelyan et al., 2019) and PyMARL [7] with StarCraft version SC2.4.6.2.69232 The architecture of all agents' policy networks is a deep recurrent Q-network (Hausknecht & Stone, 2015) consisting of two 64-dimensional fully connected layers and one 64-dimensional GRU. The mixing networks consist of a single hidden layer with 32 hidden widths and ELU activation functions. Hypernetworks consists of two layers with 64 hidden widths and ReLU activation functions. Also, the hyperparameters of WQMIX [8] (Rashid et al., 2020a) and QPLEX [9] (Wang et al., 2020a) are the same as in their GitHub codes. However, unlike they used a specific configuration for some environments, such as adding non-linearity for MMM2, we used the same setting for all environments. Finally, the hyperparameters of DFAC [10] (Sun et al., 2021) are the same as in their GitHub code. As in their paper, we use 256 hidden layer sizes for DMIX (Sun et al., 2021) in the MMM2 environment and 512 hidden layer sizes for DDN (Sun et al., 2021) in the MMM2 environment. For other scenarios, we fix all the hidden layer sizes as 64 for a fair comparison. There may be slight differences compared to their paper due to differences in StarCraft version.

Like the DFAC paper (Sun et al., 2021), following the optimizer of the IQN (Dabney et al., 2018a), we used the Adam optimizer. For other methods except for DRIMA and DFAC, according to their papers, all neural networks are trained using the RMSProp optimizer with a 0.0005 learning rate. We use $\epsilon$-greedy action selection with decreasing $\epsilon$ from 1 to 0.05 for exploration, following Samvelyan et al. (2019). For the discount factor, we set $\gamma = 0.99$. The replay buffer stores 5000 episodes at most, and the mini-batch is 32. Using a Nvidia Titan Xp graphic card, the training time varies from 8 hours to 24 hours for different scenarios.

To apply IQN for the true action-value estimator, we use an additional network, which computes an embedding $\phi(w_{\text{env}})$ for the sample point $w_{\text{env}}$. We calculate the embedding of $w_{\text{env}}$ with cosine basis functions and utilize element-wise (Hadamard) product, instead of a simple vector concatenation, for merging function as in IQN paper. The element-wise product forces a sufficiently early interaction between the two representations. The only difference is that because we use mixing networks rather than DQN, the hidden layer of the mixing network, not the convolution features, is multiplied by the embedded sample point to force interaction between the features and the embedded sample point.

For the loss function $\mathcal{L}_{\text{ub}}$, we fixed the value only for the threshold $\widehat{Q}_{\text{jt}}(s, \boldsymbol{\tau}, \boldsymbol{u}_{\text{opt}})$ of the clipping function that receives optimal actions as input. To combine our loss functions, we obtain the following objective, which is minimized in an end-to-end manner to train the true action-value estimator and the transformed action-value estimator:

$$\mathcal{L} = \mathcal{L}_{\text{td}} + \lambda_{\text{opt}}\mathcal{L}_{\text{opt}} + \lambda_{\text{nopt}}\mathcal{L}_{\text{nopt}} + \lambda_{\text{ub}}\mathcal{L}_{\text{ub}}$$

where $\lambda_{\text{opt}}, \lambda_{\text{nopt}} > 0$ are hyperparameters controlling the importance of each loss function. We set $\lambda_{\text{opt}} = 3$ and $\lambda_{\text{nopt}}, \lambda_{\text{ub}} = 1$.

As for the loss, the calculation of $\mathcal{L}_{\text{td}}$ was performed for multiple samples at the same time as in IQN. We simply set the number of samples $N_{\text{env}}$ for $w_{\text{env}}$ and $N'_{\text{env}} = 8$ for $w'_{\text{env}}$. Also, for the losses for the transformed action-value estimator, the expected value of true action-value estimator $Q_{\text{jt}}(\boldsymbol{u}) = \mathbb{E}_{w_{\text{env}}}[Z_{\text{jt}}(\boldsymbol{u}, w_{\text{env}})]$ was obtained with $K_{\text{env}} = 8$ samples. For the IQN (Dabney et al., 2018a) architecture, we use cosine basis functions with an embedding size of 64, ReLU nonlinearity, and multiplicative interaction according to their paper. For environment-wise risk-sensitivity, we used sampled $w_{\text{env}}$ from $\mathcal{U}(0, 0.25)$ as risk-averse, and $\mathcal{U}(0.75, 1.0)$ as risk-seeking,

For agent-wise risk, we don't need to learn $Z_{\text{tran}}$ values for all possible $w_{\text{agt}}$ because, unlike $Z_{\text{jt}}^{\text{target}}$, the value distribution of $Z_{\text{tran}}$ is not used for the target in loss functions. Therefore, for simplicity of implementation, we sample a fixed set of $w_{\text{agt}}$ with intervals of 0.1 from 0.1 to 1. In the execution phase, we select optimal action based on our agent-wise risk-sensitivity $w_{\text{agt}}$. For the agent-wise risk-sensitivity, we used $w_{\text{agt}} = 0.5$ as risk-neutral, $w_{\text{agt}} = 1.0$ as risk-seeking, and $w_{\text{agt}} = 0.1$ as risk-averse.

---

[6]https://github.com/oxwhirl/smac
[7]https://github.com/oxwhirl/pymarl
[8]https://github.com/oxwhirl/wqmix
[9]https://github.com/wjh720/QPLEX
[10]https://github.com/j3soon/dfac

# E  ADDITIONAL EXPERIMENTS ON THE STOCHASTIC TWO-STEP GAME

## E.1  COMPARING DRIMA WITH OTHER BASELINES

To raise understandings about DRIMA and existing works, we additionally provide results from more diverse algorithms including QMIX (Rashid et al., 2018), QTRAN (Son et al., 2019), and QPLEX (Wang et al., 2020a) on the stochastic two-step game. As shown in Table 3, one can find that (1) QMIX fails to learn the optimal behavior in the first step and (2) QTRAN and QPLEX only learn only type of risk-sensitivity (i.e., agent-wise risk seeking and environment-wise risk neutral) while DRIMA has ability to learn diverse type of risk-sensitivity. For (1), the test reward by QMIX ranges between $-0.65$ and $1.97$; it means that QMIX fails to learn the optimal behavior $(A, A)$ that has the highest payoff of $12$ in the first step. We understand that this is because QMIX has limited representational power due to the monotonic network so that they cannot represent agent-wise risk-seeking policies. For (2), while QTRAN and QPLEX achieve to learn the optimal policy due to their increased representational power, they are only capable of learning single type of risk-sensitivity; QTRAN and QPLEX have no explicit leverage to adjust risk-sensitivity. In the real world, there might exist situations that different type of risk-sensitivity is required such as agent-wise risk-seeking yet environment-wise risk-averse, so we expect that the scheme of disentangling risk sources by DRIMA guide new interesting research direction.

Table 3: Test rewards and trained policy in the stochastic two-step matrix game for DRIMA, QMIX, QTRAN, and QPLEX across twelve random seeds. Note that QMIX, QTRAN, and QPLEX have no ability to adjust risk sensitivity.

| ALGORITHM | RISK SENSITIVITY | | TEST REWARD | | | FIRST STEP | | SECOND STEP | |
|---|---|---|---|---|---|---|---|---|---|
| | AGENT | ENV | MIN | MEAN | MAX | $u_1$ | $u_2$ | $u_1$ | $u_2$ |
| DRIMA | NEUTRAL | AVERSE | -1.00 | -1.00 | -1.00 | B OR C | B OR C | B | B |
| | NEUTRAL | NEUTRAL | -1.28 | 0.89 | 4.13 | B OR C | B OR C | A | A |
| | NEUTRAL | SEEKING | -6.65 | 2.34 | 10.99 | B OR C | B OR C | C | C |
| | SEEKING | AVERSE | 7.00 | 7.00 | 7.00 | A | A | B | B |
| | SEEKING | NEUTRAL | 6.03 | 8.98 | 11.07 | A | A | A | A |
| | SEEKING | SEEKING | 2.78 | 5.50 | 13.01 | A | A | C | C |
| QMIX | | | -0.65 | 0.61 | 1.97 | B OR C | B OR C | A | A |
| QTRAN | | | 5.92 | 9.12 | 11.31 | A | A | A | A |
| QPLEX | | | 6.17 | 8.96 | 10.54 | A | A | A | A |

### E.2 SIMPLE ONE-STEP MATRIX GAME WITH NOISY AGENTS

In previous environments, agent-wise risk-seeking always performs well because assuming "cooperative" teammates in training helps to learn tactics to beat enemies. However, in real world, randomly behaving agents or adversarial teammates are likely to exist. To demonstrate the importance of an agent-wise risk-neutral policy, we increased the randomness of the agents' behavior in this experiment. In this setting, the agents act randomly with a 50% probability even during the test phase. As shown in Table 4, we eliminate the second step for environment-wise risk in the previous two-step simple matrix game, and increased the penalty for choosing the wrong action in the first step. We compare DRIMA, DMIX (Sun et al., 2021) and OW-QMIX (Rashid et al., 2020a) with varying risk-sensitivity, conducted over 50k steps. We employ a full exploration scheme (*i.e.*, $\epsilon = 1$ in $\epsilon$-greedy) so that all available states will be visited.

In this experiment, the training settings are the same as in the previous simple two-step matrix game experiment in section 4, so the agents learn the same policy as before. As shown in Table 5, we observe that risk-neutral agents outperform risk-seeking agents. This is because agent-wise risk-seeking agents always assume that other agents will be cooperative with them. These agents are easily degraded in the presence of other non-cooperating agents. In the real world applications, when a malicious external agent is added, there is a risk that multi-agent system is easily broken. Therefore, learning robust multi-agent systems against adversarial agents is an interesting future direction of research. We hope that our idea will help this new future research directions such as adversarial training in multi-agent reinforcement learning.

Table 4: Payoff matrix of the one-step game.

| $u_2$ / $u_1$ | A | B | C |
|---|---|---|---|
| A | 8 | -100 | -100 |
| B | -100 | 0 | 0 |
| C | -100 | 0 | 0 |

Table 5: Test rewards and trained policy in the one-step matrix game for DRIMA, DMIX, and OW-QMIX with varying risk-sensitivity across twelve random seeds.

| ALGORITHM | RISK SENSITIVITY | | TEST REWARD WITH NOISY AGENTS |
|---|---|---|---|
| | AGENT | ENV. | MEAN |
| DRIMA | NEUTRAL | NEUTRAL | **-26.92** |
| | SEEKING | NEUTRAL | -45.01 |
| DMIX | NEUTRAL | | -27.84 |
| | SEEKING | | -45.64 |
| OW-QMIX | NEUTRAL | | -27.52 |
| | SEEKING | | -44.48 |

# F  ADDITIONAL EXPERIMENTS ON SMAC

## F.1  ABLATION STUDIES

**DRIMA with varying risk-levels.** To investigate the effect of different risk levels in DRIMA, we conduct experiments that adjust agent-wise risk and environment-wise risk. As shown in Figure 8, we find that leveraging risk levels with disentanglement indeed affects the performance in a hard task (i.e., MMM2). Interestingly, one can note that agent-wise risk-seeking and environment-wise risk-seeking shows the strongest performance. We believe that it is critical for teammates to act cooperatively (i.e., agent-wise risk-seeking) and enhance exploration through an environment-wise risk-seeking objective in order to find the optimal tactic in such a hard task.

**Comparisons to risk-senstive variants of DDN and DMIX.** To further understand the effectiveness of disentangling risk sources, we compare DRIMA and risk-sensitive variants of DDN and DMIX in Figure 9. Although DFAC proposes only risk-neutral policies in their original paper, we additionally implement their risk-sensitive variants by adjusting the quantile-sampling range, i.e., sampling quantiles in $[0, 0.25]$ for risk-averse policy and $[0.75, 1]$ for risk-seeking policy. As shown in Figure 9, DRIMA also achieves superior performance over risk-sensitive variants of DFAC. The gain mainly comes from the ability of DRIMA to explicitly separate the two risk sources.

**Robustness of DRIMA** In Figure 10, in order to show the robustness of DRIMA, we report the performance over the second-best baseline across relatively easy tasks: 5m_vs_6m-v1, 3s_vs_5z-v1, and 8m_vs_9m-v1. We note that changing environmental-wise risks still result to DRIMA performing the best. It shows not only the empirical strength of DRIMA but also one interesting finding; environmental-risk does not matter in finding a desirable tactic in such easy tasks. However, note that in 8m_vs_9m-v1, only environment-wise risk-averse agents reached a 100% win rate, although other agents also achieve high win rates. This means that a safe policy can be beneficial for easy tasks.

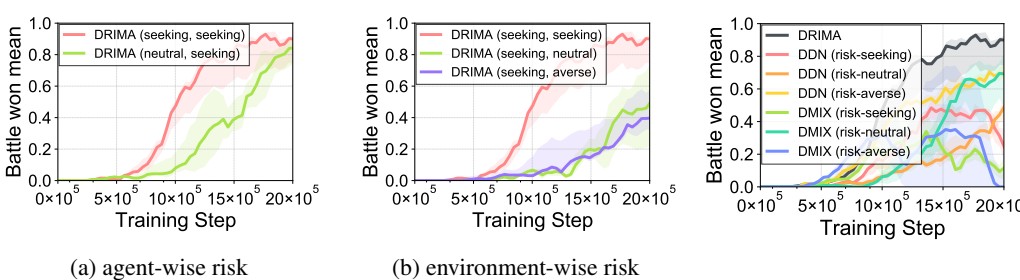

(a) agent-wise risk  (b) environment-wise risk

Figure 8: Ablation studies on varying risk-levels (agent-wise risk $w_{\mathtt{agt}}$, environment-wise risk $w_{\mathtt{env}}$) of DRIMA in MMM2-v1, one of *SUPER HARD* tasks.

Figure 9: Comparison of DRIMA to risk-sensitive variants of DDN and DMIX in MMM2-v1.

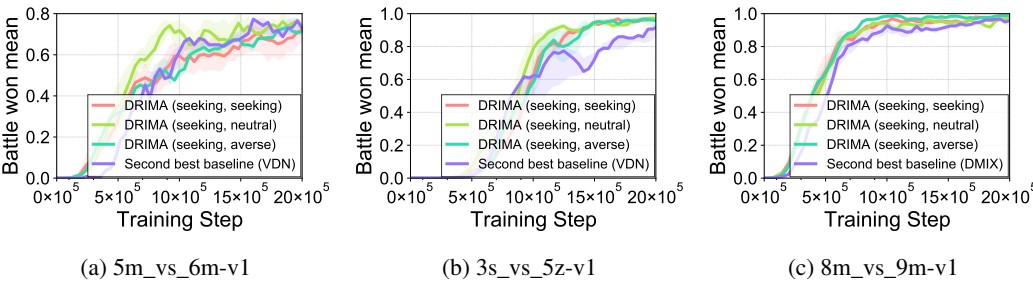

(a) 5m_vs_6m-v1  (b) 3s_vs_5z-v1  (c) 8m_vs_9m-v1

Figure 10: Robustness of DRIMA in relatively easy tasks. Differing risk-levels (agent-wise risk $w_{\mathtt{agt}}$, environment-wise risk $w_{\mathtt{env}}$) in DRIMA do not degrade below the second-best baseline.

## F.2 EXPERIMENTAL RESULTS ACROSS MORE VARIOUS MAPS

We provide experimental results across additional maps on SMAC. We report median test win rate with $25\% - 75\%$ percentile over five random seeds, comparing DRIMA with five baselines. One can observe DRIMA demonstrates competitive performance in overall.

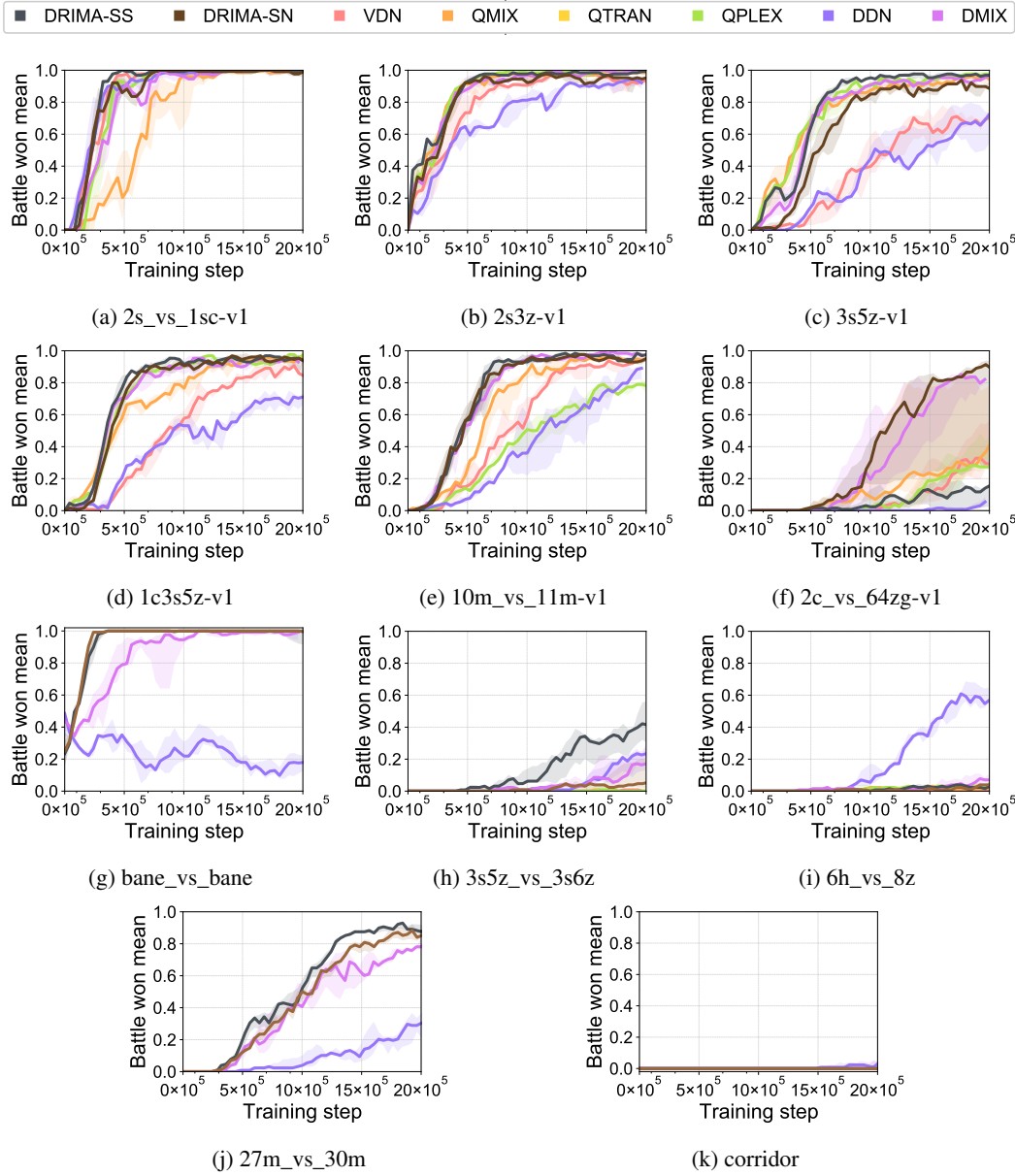

Figure 11: Median test win rate with 25%-75% percentile over four random seeds, comparing DRIMA with five baselines.

