# OpenReview forum: "Disentangling Sources of Risk for Distributional Multi-Agent Reinforcement Learning"
_ICLR.cc/2022/Conference — ICLR 2022 Submitted_

### Official Review · Reviewer_DnPu · 2021-10-30

**Correctness:** 4
**Technical Novelty And Significance:** 3
**Empirical Novelty And Significance:** 3
**Recommendation:** 6
**Confidence:** 4

**Main Review:**

# Strengths
- The performance benefits of DRIMA are clear and consistent -- at least in the SMAC testbed (which represents a challenging set of tasks).
- The authors compare against a good variety of baselines.
- The experiments include additional characterisation of DRIMA, beyond simply showing that it is SOTA. In particular, they look at how different risk choices impact performance.
- One insight is that, in the hardest setting, risk-seeking preferences are useful, both in terms of agent-based and environment-based risk. This naturally raises the question of whether risk-seeking is simply the important piece, but the authors demonstrate that risk-seeking *without disentangling risk source* is not particularly helpful. This provides a nice validation of the claim that disentangling risk is key to DRIMA's success.

# Weaknesses
- The methodology sections of the paper are very hard to follow. It often feels as though one needs a strong background with Centralized Training Decentralized Execution paradigms. In any case, I found that Section 3 left me with very little concrete understanding of how DRIMA works. However, owing to the rest of the paper being sufficiently accessible, I was still able to appreciate some of the contributions.
- For emphasis, I will reiterate that clarity of writing is the biggest weakness of the paper. One example is that I cannot connect the risk levels $w_\texttt{agt}$ and $w_{\texttt{env}}$ parameters (which I understand intuitively) to the training, execution, or design of the networks.
- The terminology use is confusing. In particular "transformed" and "true" action-values are not clearly explained, despite the amount of text they are given. There is lots of room for improvement here. The paper does not necessarily need more detail, but I think it would strongly benefit from a careful re-organization of Section 3. A general audience will not be able to understand the methodology. I am a MARL practitioner and I can't understand the methodology as written.


# Comments / Questions
- What is the best way to think of the reward, i.e. $r(s, \mathbf{u})$, versus the agent-wise utility? Is the agent-wise utility simply the estimate of the joint Q value given the observations available to the agent?
- If there is some way to visualize/inspect the way that disentangled risk affects the decision process or representation of value (compared to non-disentangled variants), that would be very interesting to see. Perhaps it is possible to compare how, for some reference state, risk is estimated in the disentangled versus non-disentangled representations. I do not think the paper strictly *needs* this analysis, but something along these lines could provide meaningful insight into what kinds of benefits DRIMA learning can create.

**Summary Of The Paper:**

This work introduces DRIMA, a Multi-Agent Reinforcement Learning algorithm that attempts to learn risk-specific behaviors, where risk is separately considered by its source: from other agents or from the environment. The authors explain their method as a novel combination of prior approaches and validate it empirically in the Starcraft Multi-Agent Challenge testbed. In addition, they provide some demonstration that non-disentangled risk sensitivity does not yield the same benefits.

**Summary Of The Review:**

My starting recommendation is: weak accept.

I think the empirical results speak for themselves, and the concept of disentangled risk is well motivated. In addition, I think the ability to separately tailor risk-preference based on the source of risk is an interesting contribution. Unfortunately, much of the contribution is hard to appreciate because various technical descriptions are vague and overly reliant on jargon.

I am confident that this paper could become a clearer "accept" if the communication were improved. In addition, some extra analyses to more deeply explore the impact of disentangling risk sources would validate the technique. I am happy to increase my score, to the extent that these suggestions are followed.

---

> ### Author Response · Authors · 2021-11-20
> **Response to Reviewer DnPu (1)**
>
> We express our deep appreciation for your time and insightful comments. We address your comments one by one. The revisions made are marked with “$\text{\color{magenta}magenta}$” in the revised paper.
>
> **Q1. The methodology sections of the paper are very hard to follow. I will reiterate that clarity of writing is the biggest weakness of the paper.**
>
> **A1.** Thank you for the helpful suggestion. To improve readability, we reorganized the preliminary and methodology sections. In particular, we added Section 3.1 at the beginning of the method section to further improve understanding of DRIMA. Also, we largely updated Section 2.1 and Section 4 to help a broader audience understand our work.
>
> ---
>
> **Q2. The terminology usage is confusing. In particular “transformed” and “true” action-values are not clearly explained, despite the amount of text they are given.**
>
> **A2.** Our terminology usage largely follows the QTRAN paper.  To be specific, the true joint action-value refers to the estimator learned via TD-error which has complete information about the global state. Meanwhile, the transformed action-value denotes the estimator that satisfies both desired properties: (1) its optimal actions are consistent with the “true” action-value estimator, and (2) it can be decentralized into agent-wise utility functions.  In our revised draft, we have further clarified this in Sections 2.1, 3.1, and 3.2.
>
> ---
>
> **Q3. What is the best way to think of the reward, versus the agent-wise utility?**
>
> **A3.** They are different because the value of the agent-wise utility can be any value as long as the argmax action of the agent-wise utility and the joint action-value is the same. Conceptually, an agent-wise utility is similar to an actor in an actor-critic method. This is why we use the word “utility” rather than agent-wise “action-value”. In our revised draft, we further clarified this in Section 3.3.

---

> > ### Author Response · Authors · 2021-11-20
> > **Response to Reviewer DnPu (2)**
> >
> > **Q4. If there is some way to visualize/inspect the way that disentangled risk affects the decision process or representation of value, that would be very interesting to see.**
> >
> > **A4.** Thank you for the suggestion. To further inspect how the agent-wise risk level $w_{\mathtt{agt}}$ and environment-wise risk level $w_{\mathtt{env}}$ influence the behavior of agents, we evaluate DRIMA and baselines in a stochastic two-step matrix game (see Section 4 of our revised draft). In this game, two agents select an action in {$\{A, B, C\}$} and receive a shared global payoff (reward) as Tables 1 and 2 below show. The payoffs of the first and the second step are designed to assess whether an algorithm is able to handle agent-wise and environment-wise risks, respectively. For the first step, agent-wise risk-seeking (cooperation) is crucial because the optimal action is $(A, A)$ but taking action $A$ is very risky because when the teammate chooses to be non-cooperative (i.e., by selecting action $B$ or $C$), a catastrophic reward of -12 is provided.. In the second step, handling environment-wise risks is highlighted since there are various combinations of mean and variance depending on joint actions, i.e., $\mathcal{N}(-1, 10)$ for the action $(C, C)$ and $\mathcal{N}(-1, 0)$ for the action $(B, B)$.
> >
> > As shown in Table 3 below, we observe that DRIMA is able to adjust agent-wise and environment-wise risk separately so that it can achieve the optimal policy that requires agent-wise risk-seeking and environment-wise risk-neutral policy (policy whose mean test reward is maximum). Note that other baselines including DMIX, Weighted QMIX, QMIX, QTRAN, and QPLEX have a limitation in adjusting risk-sensitivity, so they fail to represent the optimal policy.
> >
> > We provide more environmental details and analysis in Section 4 in our revised draft.
> >
> > ###### $\textbf{Table1: Payoff in the first step of the two-step stochastic matrix game}$
> > \begin{array}{cc|ccc}
> > & &    & u_{2} & \newline
> > & & A & B & C \newline
> > \hline
> > & A & \mathcal{N}(8, \textcolor{gray}{0}) & \mathcal{N}(-12, \textcolor{gray}{0}) & \mathcal{N}(-12, \textcolor{gray}{0}) \newline
> > u_{1} & B & \mathcal{N}(-12, \textcolor{gray}{0})  & \mathcal{N}(0, \textcolor{gray}{0}) & \mathcal{N}(0, \textcolor{gray}{0}) \newline
> > & C & \mathcal{N}(-12, \textcolor{gray}{0})  & \mathcal{N}(0, \textcolor{gray}{0}) & \mathcal{N}(0, \textcolor{gray}{0}) \newline
> > \end{array}
> >
> > ###### $\textbf{Table2: Payoff in the second step of the two-step stochastic matrix game}$
> > \begin{array}{cc|ccc}
> > & &    & u_{2} & \newline
> > & & A & B & C \newline
> > \hline
> > & A & \mathcal{N}(1, 5) & \mathcal{N}(0, 5) & \mathcal{N}(0, 5) \newline
> > u_{1} & B & \mathcal{N}(0, 5)  & \mathcal{N}(-1, 0) & \mathcal{N}(-1, 5) \newline
> > & C & \mathcal{N}(0, 5)  & \mathcal{N}(-1, 5) & \mathcal{N}(-1, 10) \newline
> > \end{array}
> >
> >
> > ###### $\textbf{Table3: Test rewards and trained policy in the stochastic two-step matrix game}$
> > \begin{array}{l|ccc cc cc}
> > \text{Algorithm} & & \text{Test reward} & & & & & & \newline
> > \text{(agent-wise risk, env-wise risk)} & \text{Min} & \text{Mean} & \text{Max} & 1^{\text{st}} ~ u_{1}& 1^{\text{st}} ~ u_{2} & 2^{\text{nd}} ~ u_{1} & 2^{\text{nd}} ~ u_{2}\newline
> > \hline
> > \text{DRIMA (Neutral, Averse)} & -1.00 & -1.00 & -1.00 & \text{B or C} & \text{B or C} & \text{B} & \text{B} & \newline
> > \text{DRIMA (Neutral, Neutral)} & -1.28 & 0.89 & 4.13 & \text{B or C} & \text{B or C} & \text{A} & \text{A} \newline
> > \text{DRIMA (Neutral, Seeking)} & -6.65 & 2.34 & 10.99 & \text{B or C} & \text{B or C} & \text{C} & \text{C} \newline
> > \text{DRIMA (Seeking, Averse)} & \textbf{7.00} & 7.00 & 7.00 & \text{A} & \text{A} & \text{B} & \text{B} \newline
> > \text{DRIMA (Seeking, Neutral)} & 6.03 & \textbf{8.98} & 11.07 & \text{A} & \text{A} & \text{A} & \text{A} \newline
> > \text{DRIMA (Seeking, Seeking)} & 2.78 & 5.50 & \textbf{13.01} & \text{A} & \text{A} & \text{C} & \text{C} \newline
> > \hline
> > \text{DMIX (Averse)} & -1.00 & -1.00 & -1.00 & \text{B or C} & \text{B or C} & \text{B} & \text{B} \newline
> > \text{DMIX (Neutral)} & -2.66 & 0.89 & 2.94 & \text{B or C} & \text{B or C} & \text{A} & \text{A}  \newline
> > \text{DMIX (Seeking)} & 1.89 & 6.96 & \textbf{12.81} & \text{A} & \text{A} & \text{C} & \text{C} \newline
> > \hline
> > \text{OW-QMIX (Neutral)} & -3.04 & 0.63 & 2.98 & \text{B or C} & \text{B or C} & \text{A} & \text{A} \newline
> > \text{OW-QMIX (Seeking)} & 1.95 & 6.94 & \textbf{12.87} & \text{A} & \text{A} & \text{C} & \text{C}  \newline
> > \hline
> > \text{QMIX (-)}& -0.65 & 0.61 & 1.97 & \text{B or C} & \text{B or C} & \text{A} & \text{A} \newline
> > \text{QTRAN (-)} & 5.92 & 9.12 & 11.31 & \text{A} & \text{A} & \text{A} & \text{A} \newline
> > \text{QPLEX (-)} & 6.17 & 8.96 & 10.54 & \text{A} & \text{A} & \text{A} & \text{A} \newline
> > \end{array}

---

> > > ### Comment · Reviewer_DnPu · 2021-11-22
> > > **Response**
> > >
> > > Thank you for your responses and for your efforts to incorporate my feedback. I feel as though my criteria for improvement have been addressed.
> > >
> > > I am inclined to consider a higher score (8), but before finalizing anything, I would like to leave time for discussion and input from other reviewers. In any case, my overall recommendation is certainly on the side of accept.

---

> > > > ### Author Response · Authors · 2021-11-30
> > > > **We highly appreciate your insightful and helpful comments.**
> > > >
> > > > We highly appreciate your insightful and helpful comments. We are glad to hear that our response helped you understand our method.

---

### Official Review · Reviewer_LvSz · 2021-11-01

**Correctness:** 2
**Technical Novelty And Significance:** 2
**Empirical Novelty And Significance:** 2
**Recommendation:** 3
**Confidence:** 4

**Main Review:**


The paper describes a few different loss functions that try to combine
distributional RL with decentralised reinforcement learning in
POMDPs. However, it does not demonstrate theoretically that these loss
functions achieve the intended effect, i.e. that they are a natural
outcome of the problem of decentralised distributed RL.  As such, the
paper is purely experimental.

The discussion of QTRAN and weighted QMIX is a bit peculiar. The
authors regard them as risk-seeking, while this is not explicit in the
cited papers. They draw this conclusion based on how the loss function
is defined, with OW-QMIX being 'optimstic'. But this is, to me, only
an intuitive understanding of the algorithms and does not arise from
them actually performing a risk-sensitive maximisation.

It is not clear to me why explicitly separating the two risk sources
helps in this scenario, or how it is achieved.  The authors mention
the sign of the TD-error as a way to obtain environment risk, but they
do not show that this is a reasonable idea.

So, viewing everything as purely heuristic, let us look at the
experimental results. In terms of the 'battle won mean' metric,
performance is competitive with the state of the art.

Although the paper initially makes much of the problem of
disentangling risk, this is only analysed in a short paragraph in the
experiments.

Note:
You say that the assumption is $\max_u Q_{jt}(s, \tau, u) = [\max_{u_i} q_i(\tau_i, u_i)$.
I generally fail to see how this can be the case,since $q_i$ is marginalising over all possible states.
I note that in (2), there is no $s$ in $Q_{jt}$.


**Summary Of The Paper:**

This paper describes a few different loss functions and heuristics to use
distributional RL in Dec-POMDPs in order to obtain risk-sensitive policies.

**Summary Of The Review:**

The heuristics are poorly motivated theoretically, and the experiments do not clearly highlight how disentangling risk sources can be helpful to drive risk-sensitive behaviour.

---

> ### Author Response · Authors · 2021-11-20
> **Response to Reviewer LvSz (1)**
>
> We express our deep appreciation for your time and insightful comments. We address your comments one by one. The revisions made are marked with “$\text{\color{magenta}magenta}$” in the revised paper.
>
> **Q1. The paper is purely experimental. It is not clear to me why explicitly separating the two risk sources helps in this scenario.**
>
> **A1.** To further investigate how the agent-wise risk level $w_{\mathtt{agt}}$ and environment-wise risk level $w_{\mathtt{env}}$ influence the behavior of agents, we evaluate DRIMA and baselines in a stochastic two-step matrix game (see Section 4 of the revised draft). In this game, two agents select an action in {$\{A, B, C\}$} and receive a shared global payoff (reward) as Tables 1 and 2 show. The payoffs of the first and the second step are designed to assess whether an algorithm is able to handle agent-wise and environment-wise risks, respectively. For the first step, agent-wise risk-seeking (cooperation) is crucial because the optimal action is $(A, A)$ but taking action $A$ is very risky because when the teammate chooses to be non-cooperative (i.e., by selecting action $B$ or $C$), a catastrophic reward of -12 is provided. In the second step, handling environment-wise risks is highlighted since there are various combinations of mean and variance depending on joint actions, i.e., $\mathcal{N}(-1, 10)$ for the action $(C, C)$ and $\mathcal{N}(-1, 0)$ for the action $(B, B)$.
>
> As shown in Table 3, we observe that DRIMA is able to adjust agent-wise and environment-wise risk separately so that it can achieve the optimal policy that requires agent-wise risk-seeking and environment-wise risk-neutral policy (policy whose mean test reward is maximum). Note that other baselines including DMIX, Weighted QMIX, QMIX, QTRAN, and QPLEX have a limitation in adjusting risk-sensitivity, so they fail to represent the optimal policy.
>
> We provide more environmental details and analysis in Section 4 of our revised draft.
>
> ###### $\textbf{Table1: Payoff in the first step of the two-step stochastic matrix game}$
> \begin{array}{cc|ccc}
> & &    & u_{2} & \newline
> & & A & B & C \newline
> \hline
> & A & \mathcal{N}(8, \textcolor{gray}{0}) & \mathcal{N}(-12, \textcolor{gray}{0}) & \mathcal{N}(-12, \textcolor{gray}{0}) \newline
> u_{1} & B & \mathcal{N}(-12, \textcolor{gray}{0})  & \mathcal{N}(0, \textcolor{gray}{0}) & \mathcal{N}(0, \textcolor{gray}{0}) \newline
> & C & \mathcal{N}(-12, \textcolor{gray}{0})  & \mathcal{N}(0, \textcolor{gray}{0}) & \mathcal{N}(0, \textcolor{gray}{0}) \newline
> \end{array}
>
> ###### $\textbf{Table2: Payoff in the second step of the two-step stochastic matrix game}$
> \begin{array}{cc|ccc}
> & &    & u_{2} & \newline
> & & A & B & C \newline
> \hline
> & A & \mathcal{N}(1, 5) & \mathcal{N}(0, 5) & \mathcal{N}(0, 5) \newline
> u_{1} & B & \mathcal{N}(0, 5)  & \mathcal{N}(-1, 0) & \mathcal{N}(-1, 5) \newline
> & C & \mathcal{N}(0, 5)  & \mathcal{N}(-1, 5) & \mathcal{N}(-1, 10) \newline
> \end{array}
>
> ###### $\textbf{Table3: Test rewards and trained policy in the stochastic two-step matrix game}$
> \begin{array}{l|ccc cc cc}
> \text{Algorithm} & & \text{Test reward} & & & & & & \newline
> \text{(agent-wise risk, env-wise risk)} & \text{Min} & \text{Mean} & \text{Max} & 1^{\text{st}} ~ u_{1}& 1^{\text{st}} ~ u_{2} & 2^{\text{nd}} ~ u_{1} & 2^{\text{nd}} ~ u_{2}\newline
> \hline
> \text{DRIMA (Neutral, Averse)} & -1.00 & -1.00 & -1.00 & \text{B or C} & \text{B or C} & \text{B} & \text{B} & \newline
> \text{DRIMA (Neutral, Neutral)} & -1.28 & 0.89 & 4.13 & \text{B or C} & \text{B or C} & \text{A} & \text{A} \newline
> \text{DRIMA (Neutral, Seeking)} & -6.65 & 2.34 & 10.99 & \text{B or C} & \text{B or C} & \text{C} & \text{C} \newline
> \text{DRIMA (Seeking, Averse)} & \textbf{7.00} & 7.00 & 7.00 & \text{A} & \text{A} & \text{B} & \text{B} \newline
> \text{DRIMA (Seeking, Neutral)} & 6.03 & \textbf{8.98} & 11.07 & \text{A} & \text{A} & \text{A} & \text{A} \newline
> \text{DRIMA (Seeking, Seeking)} & 2.78 & 5.50 & \textbf{13.01} & \text{A} & \text{A} & \text{C} & \text{C} \newline
> \hline
> \text{DMIX (Averse)} & -1.00 & -1.00 & -1.00 & \text{B or C} & \text{B or C} & \text{B} & \text{B} \newline
> \text{DMIX (Neutral)} & -2.66 & 0.89 & 2.94 & \text{B or C} & \text{B or C} & \text{A} & \text{A}  \newline
> \text{DMIX (Seeking)} & 1.89 & 6.96 & \textbf{12.81} & \text{A} & \text{A} & \text{C} & \text{C} \newline
> \hline
> \text{OW-QMIX (Neutral)} & -3.04 & 0.63 & 2.98 & \text{B or C} & \text{B or C} & \text{A} & \text{A} \newline
> \text{OW-QMIX (Seeking)} & 1.95 & 6.94 & \textbf{12.87} & \text{A} & \text{A} & \text{C} & \text{C}  \newline
> \hline
> \text{QMIX (-)}& -0.65 & 0.61 & 1.97 & \text{B or C} & \text{B or C} & \text{A} & \text{A} \newline
> \text{QTRAN (-)} & 5.92 & 9.12 & 11.31 & \text{A} & \text{A} & \text{A} & \text{A} \newline
> \text{QPLEX (-)} & 6.17 & 8.96 & 10.54 & \text{A} & \text{A} & \text{A} & \text{A} \newline
> \end{array}

---

> > ### Author Response · Authors · 2021-11-20
> > **Response to Reviewer LvSz (2)**
> >
> > **Q2. The discussion of OW-QMIX is a bit peculiar.**
> >
> > **A2.** Thanks for your kind comment. We agree that describing OW-QMIX as agent-wise risk can be confusing. As written in Sections 2 and 3, Weighted QMIX is a paper that applies optimistic training methods such as hysteretic Q-learning [1] to the CTDE paradigm. The relation between optimistic training and distributional reinforcement learning has already been theoretically analyzed in other papers [2, 3]. In our revised draft, we have further clarified this in Sections 2 and 3.
> >
> > **Reference**
> >
> > [1] Matignon, Laëtitia, Guillaume J. Laurent, and Nadine Le Fort-Piat. "Hysteretic q-learning: an algorithm for decentralized reinforcement learning in cooperative multi-agent teams." 2007 IEEE/RSJ International Conference on Intelligent Robots and Systems. IEEE, 2007.
> >
> > [2] Rowland, Mark, et al. "Temporal Difference and Return Optimism in Cooperative Multi-Agent Reinforcement Learning."
> >
> > [3] Lyu, Xueguang, and Christopher Amato. "Likelihood quantile networks for coordinating multi-agent reinforcement learning." arXiv preprint arXiv:1812.06319 (2018).
> >
> > ---
> >
> > **Q3. I generally fail to see how decentralizable action-value estimator assumption can be the case.**
> >
> > **A3.** As we explained in Section 2.1, the conditions for decentralization are satisfied by various MARL papers such as VDN, QMIX, and QTRAN in their own way. First of all, the reason this condition is proposed is that multi-agent reinforcement learning includes exponential growth of the action space in the number of agents. Therefore, we need an assumption to extract the optimal action through only individual utilities rather than the joint action-value estimator.
> >
> > The reason why the state does not exist in Equation (2) is that their paper does not use (global) state information for the agent-wise utility function in the literature. Namely, only partial observation is given for the agent in the execution phase with inaccessibility to the (global) state, especially in SMAC which is a widely-used benchmark in the MARL community. Nevertheless, conceptually, their method can be modified by adding state, but we used the equation written in their original paper as it is.

---

> > > ### Comment · Reviewer_LvSz · 2021-11-21
> > > **TO-DO**
> > >
> > > Thank you for the responses and the clarification. I still think that this heuristic and vague risk definition is insufficient.

---

> > > > ### Author Response · Authors · 2021-11-23
> > > > **Response to Reviewer LvSz (3)**
> > > >
> > > > We express our deep appreciation for your time and insightful comments. The revisions made are marked with “$\text{\color{magenta} magenta}$” in the revised paper.
> > > >
> > > > **Q1. I still think that this heuristic is insufficient.**
> > > >
> > > > **A1.**  We formally show that DRIMA can separate the two risk sources through theoretical connections between Weighted QMIX and DRIMA with optimism.
> > > >
> > > > To investigate similarities and differences between DRIMA and OW-QMIX to handle risk-sensitivity, we employ a one-step single-state matrix game. In this game, there is only one state, and all episodes finish after a single time-step. We find a connection between the DRIMA and OW-QMIX via following proposition:
> > > >
> > > > *Consider a deterministic-reward environment with a single state that terminated immediately. Then the loss ${L_{\mathtt{nopt}}}$ of DRIMA with parameters $w_\mathtt{agt}$ is equivalent to the Weighted QMIX with $\alpha = 2 * (1 - w_\mathtt{agt})$.*
> > > >
> > > >
> > > > We prove this proposition in Appendix C.3 in our revised draft. This proposition says that DRIMA and OW-QMIX become equivalent in a deterministic-reward environment. However, if the environment is highly stochastic, then optimistic algorithms can induce misplaced optimism towards uncontrollable environment dynamics, leading to suboptimal behavior. The optimistic MARL approaches ignore low returns, which are assumed to be caused by teammates' exploratory actions. This causes severe overestimation of action-values in stochastic domains. Since reward and next state in the TD-target $y$ in Weighted QMIX include randomness of the environment, agent-wise risk cannot be extracted separately enough from the sign of the TD-error. Therefore, in DRIMA, we do not use the target $y$ as done in Weighted QMIX. Instead, we use $\mathbb{E_{w_{\mathtt{env}}}}[Z_\mathtt{jt}(w_{\mathtt{env}})]$ to exclude the randomness of the environment in the target. The true action-value estimator $Z_\mathtt{jt}$ of DRIMA does not have any structural restrictions such as monotonicity in value factorization methods. Therefore, $Z_\mathtt{jt}$ can accurately represent only the randomness of the environment. We highlight that the Weighted QMIX cannot disentangle risk sources in stochastic-reward environments where environment-wise randomness exists as our experimental results in Section 4 and Appendix E.2 demonstrate.
> > > >
> > > > **Q2.  I still think that vague risk definition is insufficient.**
> > > >
> > > > **A2.** To describe risks more formally, one can formulate random variable $Z_{\mathtt{env}}^{\pi}(s, u)$ and $Z_{\mathtt{agt}}(s, u)$ that model environment-wise randomness and agent-wise randomness, respectively. The environment-wise random variable $Z_{\mathtt{env}}^{\pi}(s, u)$ is the sum of future discounted reward given “deterministic” agent-wise policy $\pi$ as follows:
> > > >
> > > > $Z_{\mathtt{env}}^{\pi}(s, u) = \Sigma_{t \geq 0} \gamma^t r(s_t,u_t)|_\pi$,
> > > >
> > > > where $\pi$ is a given deterministic policy. The random variable $Z_{\mathtt{env}}$ depends on immediate stochastic reward and stochastic state transition dynamics, which are “environmental”. Note that agent-wise randomness does not exist in the $Z_{\mathtt{env}}$ due to the given deterministic agent-wise policy.
> > > >
> > > > Meanwhile, the agent-wise random variable, which exists in MARL, implies how the other agents behave for myself to achieve high global reward as follows.
> > > >
> > > > $Z_{\mathtt{agt}}(s, u) = \arg \min_{Z’}  \mathbb{E_{\tau \sim \pi}} \[w(s,u) * (Z'(s, u) - Z_{\mathtt{env}}^{\pi}(s, u))^2\]$,
> > > >
> > > > where $\tau$ is a sampled trajectory and $w(s,u)$ is a weight function depending on agent-wise risk-level. The weight function adjusts agent-wise risk-sensitivity by selecting samples whose agent-wise random variable $Z_{\mathtt{agt}}(s, u)$ are close to the environment-wise random variable $Z_{\mathtt{env}}^{\pi}(s, u)$.
> > > > For example, hysteretic Q-learning, Weighted QMIX, and DRIMA assign a higher weighting to joint actions that could be the true optimal actions for agent-wise optimism.

---

### Official Review · Reviewer_opi1 · 2021-11-02

**Correctness:** 3
**Technical Novelty And Significance:** 2
**Empirical Novelty And Significance:** 2
**Recommendation:** 3
**Confidence:** 4

**Details Of Ethics Concerns:**

The reviewer does not see obvious ethics concerns.

**Main Review:**

***Soundness***

The main concern of the reviewer is about the soundness of the proposed method. To be specific, the reviewer is not convinced that this method can disentangle the environment- and agent-wise risk.

a) Why does $w_{env}$ reflect environment-wise risk? Conditioning the joint action-value function on $w_{env}$ is not enough. The joint value function learns the Q function for the whole team and involves both environmental randomness and other agents' uncertainty. The authors may would like to consider defining agent-wise and environment-wise uncertainty formally.

b) Similarly, why does $w_{agt}$ reflect agent-wise risk? (b-1) The difference between the transformed action-value function and joint action-value function lies in their representational capacity. (b-2) The loss function for the transformed action function is more like a soft version of the QPLEX objective. Therefore, the reviewer thinks that $w_{agt}$ cannot reflect agent-wise risk, from the perspective of loss function and NN structure.

c) Current experiments do not support these claims either. In Fig. 5, the authors show that changing $w_{agt}$ and $w_{env}$ influences performance on MMM2, but this map is not enough as far as the reviewer is concerned. According to Fig. 7,  these risk levels have limited influence on other maps. Moreover, it is _very_ important but is absent from the paper how these risk levels influence the behavior of agents. The reviewer would suggest using a grid-world example and ablating $w_{agt}$ and $w_{env}$ separately, and then observing agents' behavior.

***Evaluation***

a) The reviewer can not tell whether the results are cherry-picked. 8m_vs_9m is not in the latest version of SMAC. Please show the results (comparisons against baselines) on all 14 SMAC maps.

***Clarity***

This paper is not well written, and there are various grammatical errors throughout the paper. Some notations are not clear. For example, L_{td-IQN} is used twice (page 4 and 5) in the paper but stands for different loss functions.

**Summary Of The Paper:**

This paper introduces implicit quantile networks (IQN) into value-decomposition multi-agent reinforcement learning methods (QPLEX or weighted QMIX). The proposed method uses IQN for two components of QPLEX, for the joint action-value function and the transformed action-value function, respectively. The author expects IQN for the joint value function to account for environment-wise risk while the IQN for the transformed value function to account for agent-wise risk.

**Summary Of The Review:**

The reviewer finds the problem under research in this paper interesting. However, the reviewer is not convinced that current methods can solve the problems and disentangle the environment- and agent-wise risk. Current experiments do not resolve this concern.

---

> ### Author Response · Authors · 2021-11-20
> **Response to Reviewer opi3 (1)**
>
> We express our deep appreciation for your time and insightful comments. We address your comments one by one. The revisions made are marked with “$\text{\color{magenta}magenta}$” in the revised paper.
>
> **Q1. The authors may would like to consider defining agent-wise and environment-wise uncertainty formally**
>
> **A1.** Environment-wise randomness refers to the randomness of the reward and the next transition that can occur when the joint action is determined. If the joint action-value is unconstrained, distributional reinforcement learning can only learn environment-wise uncertainty. On the other hand, if action-values are learned through limited networks such as independent learners or a monotonic network, such constrained networks should learn the action-value by treating the actions of other agents as random. The learned policy changes according to this agent-wise randomness. For example, Independent Q-learning [1] and QMIX are agent-wise risk-neutral learning methods, and hysteretic Q-learning [2] and Weighted QMIX are agent-wise risk-seeking methods.
>
> **Reference**
>
> [1] Tan, Ming. "Multi-agent reinforcement learning: Independent vs. cooperative agents." Proceedings of the tenth international conference on machine learning. 1993.
>
> [2] Matignon, Laëtitia, Guillaume J. Laurent, and Nadine Le Fort-Piat. "Hysteretic q-learning: an algorithm for decentralized reinforcement learning in cooperative multi-agent teams." 2007 IEEE/RSJ International Conference on Intelligent Robots and Systems. IEEE, 2007.
>
> ---
>
> **Q2. It is very important but is absent from the paper how these risk levels influence the behavior of agents**
>
> **A2.** Thank you for the suggestion. To further investigate how the agent-wise risk level $w_{\mathtt{agt}}$ and environment-wise risk level $w_{\mathtt{env}}$ influence the behavior of agents, we evaluate DRIMA and baselines in a stochastic two-step matrix game in the revised draft (see Section 4). In this game, two agents select action in {$\{A, B, C\}$} and receive a shared global payoff (reward) as Tables 1 and 2 below show. The payoffs of the first and the second step are designed to assess whether an algorithm is able to handle agent-wise and environment-wise risks, respectively. For the first step, agent-wise risk-seeking (cooperation) is crucial because the optimal action is $(A, A)$ but taking action $A$ is very risky because when the teammate chooses to be non-cooperative (i.e., by selecting action $B$ or $C$), a catastrophic reward of -12 is provided. In the second step, handling environment-wise risks is highlighted since there are various combinations of mean and variance depending on joint actions, i.e., $\mathcal{N}(-1, 10)$ for the action $(C, C)$ and $\mathcal{N}(-1, 0)$ for the action $(B, B)$.
>
> As shown in Table 3 below, we observe that DRIMA is able to adjust agent-wise and environment-wise risk separately so that it can achieve the optimal policy that requires agent-wise risk-seeking and environment-wise risk-neutral policy (policy whose mean test reward is maximum). Note that other baselines including DMIX, Weighted QMIX, QMIX, QTRAN, and QPLEX have a limitation in adjusting risk-sensitivity, so they fail to represent the optimal policy.
>
> We provide more environmental details and analysis in Section 4 of our revised draft.

---

> > ### Author Response · Authors · 2021-11-20
> > **Response to Reviewer opi3 (2)**
> >
> > ###### $\textbf{Table1: Payoff in the first step of the two-step stochastic matrix game}$
> > \begin{array}{cc|ccc}
> > & &    & u_{2} & \newline
> > & & A & B & C \newline
> > \hline
> > & A & \mathcal{N}(8, \textcolor{gray}{0}) & \mathcal{N}(-12, \textcolor{gray}{0}) & \mathcal{N}(-12, \textcolor{gray}{0}) \newline
> > u_{1} & B & \mathcal{N}(-12, \textcolor{gray}{0})  & \mathcal{N}(0, \textcolor{gray}{0}) & \mathcal{N}(0, \textcolor{gray}{0}) \newline
> > & C & \mathcal{N}(-12, \textcolor{gray}{0})  & \mathcal{N}(0, \textcolor{gray}{0}) & \mathcal{N}(0, \textcolor{gray}{0}) \newline
> > \end{array}
> >
> > ###### $\textbf{Table2: Payoff in the second step of the two-step stochastic matrix game}$
> > \begin{array}{cc|ccc}
> > & &    & u_{2} & \newline
> > & & A & B & C \newline
> > \hline
> > & A & \mathcal{N}(1, 5) & \mathcal{N}(0, 5) & \mathcal{N}(0, 5) \newline
> > u_{1} & B & \mathcal{N}(0, 5)  & \mathcal{N}(-1, 0) & \mathcal{N}(-1, 5) \newline
> > & C & \mathcal{N}(0, 5)  & \mathcal{N}(-1, 5) & \mathcal{N}(-1, 10) \newline
> > \end{array}
> >
> > ###### $\textbf{Table3: Test rewards and trained policy in the stochastic two-step matrix game}$
> > \begin{array}{l|ccc cc cc}
> > \text{Algorithm} & & \text{Test reward} & & & & & & \newline
> > \text{(agent-wise risk, env-wise risk)} & \text{Min} & \text{Mean} & \text{Max} & 1^{\text{st}} ~ u_{1}& 1^{\text{st}} ~ u_{2} & 2^{\text{nd}} ~ u_{1} & 2^{\text{nd}} ~ u_{2}\newline
> > \hline
> > \text{DRIMA (Neutral, Averse)} & -1.00 & -1.00 & -1.00 & \text{B or C} & \text{B or C} & \text{B} & \text{B} & \newline
> > \text{DRIMA (Neutral, Neutral)} & -1.28 & 0.89 & 4.13 & \text{B or C} & \text{B or C} & \text{A} & \text{A} \newline
> > \text{DRIMA (Neutral, Seeking)} & -6.65 & 2.34 & 10.99 & \text{B or C} & \text{B or C} & \text{C} & \text{C} \newline
> > \text{DRIMA (Seeking, Averse)} & \textbf{7.00} & 7.00 & 7.00 & \text{A} & \text{A} & \text{B} & \text{B} \newline
> > \text{DRIMA (Seeking, Neutral)} & 6.03 & \textbf{8.98} & 11.07 & \text{A} & \text{A} & \text{A} & \text{A} \newline
> > \text{DRIMA (Seeking, Seeking)} & 2.78 & 5.50 & \textbf{13.01} & \text{A} & \text{A} & \text{C} & \text{C} \newline
> > \hline
> > \text{DMIX (Averse)} & -1.00 & -1.00 & -1.00 & \text{B or C} & \text{B or C} & \text{B} & \text{B} \newline
> > \text{DMIX (Neutral)} & -2.66 & 0.89 & 2.94 & \text{B or C} & \text{B or C} & \text{A} & \text{A}  \newline
> > \text{DMIX (Seeking)} & 1.89 & 6.96 & \textbf{12.81} & \text{A} & \text{A} & \text{C} & \text{C} \newline
> > \hline
> > \text{OW-QMIX (Neutral)} & -3.04 & 0.63 & 2.98 & \text{B or C} & \text{B or C} & \text{A} & \text{A} \newline
> > \text{OW-QMIX (Seeking)} & 1.95 & 6.94 & \textbf{12.87} & \text{A} & \text{A} & \text{C} & \text{C}  \newline
> > \hline
> > \text{QMIX (-)}& -0.65 & 0.61 & 1.97 & \text{B or C} & \text{B or C} & \text{A} & \text{A} \newline
> > \text{QTRAN (-)} & 5.92 & 9.12 & 11.31 & \text{A} & \text{A} & \text{A} & \text{A} \newline
> > \text{QPLEX (-)} & 6.17 & 8.96 & 10.54 & \text{A} & \text{A} & \text{A} & \text{A} \newline
> > \end{array}

---

> > > ### Author Response · Authors · 2021-11-20
> > > **Response to Reviewer opi3 (3)**
> > >
> > > **Q3. The reviewer can not tell whether the results are cherry-picked.**
> > >
> > > **A3.** Thank you for the suggestion. To address your concern, we conduct our experiments on all 14 SMAC maps you mentioned. As shown in tables below, we observe that DRIMA achieves state-of-the-art performance on 12 out of 14 maps. Note that DRIMA-SS and DRIMA-SN mean DRIMA (agent-wise risk-**seeking**, environment-wise risk-**seeking**) and DRIMA (agent-wise risk-**seeking**, environment-wise risk-**neutral**), respectively. One can further improve performance of DRIMA since we did not search our hyperparameters exhaustively; note that we employ the best hyperparameters (e.g., hidden dimension of agent-wise utility functions) reported by original DFAC paper on 3s5z_vs_3s6z-v1, 6h_vs_8z-v1, 27m_vs_30m-v1, corridor-v1, and MMM2-v1, but we did not search the best hidden dimension for DRIMA. We remark that DRIMA achieves competitive performance over other baselines (VDN, QMIX, and QPLEX) that have exactly fair condition with DRIMA on all the 14 maps.
> > >
> > > We are now running VDN, QMIX, and QPLEX on bane_vs_bane-v1, 27m_vs_30m-v1 and corridor-v1, and we will report their performance as soon as possible. Moreover, we are going to upload an aggregated result on 14 Maps of DRIMA and the 5 baselines.
> > >
> > >
> > > ###### $\bf{2s\\_vs\\_1sc-v1}$
> > > \begin{array}{l| cccccc}
> > > \text{Timesteps}  & 2.5 * 10^5 & 5.0 * 10^5 & 7.5 * 10^5 & 10.0 * 10^5 & 15.0 * 10^5 & 20.0 * 10^5 \newline
> > > \hline
> > > \textbf{DRIMA-SS} & 0.54 & \bf{1.00} & \bf{0.99} & 0.99 & 0.99 & \bf{1.00} \newline
> > > \textbf{DRIMA-SN} & \bf{0.61} & 0.86 & \bf{0.99} & \bf{1.00} & 0.99 & 0.99 \newline
> > > \text{VDN} & 0.55 & 0.97 & \bf{0.99} & 0.99 & 1.00 & 0.99 \newline
> > > \text{QMIX} & 0.07 & 0.20 & 0.87 & 0.97 & 0.98 & 0. 98 \newline
> > > \text{QPLEX} & 0.34 & 0.93 & 0.96 & 0.99 & \bf{1.00} & 0.99 \newline
> > > \text{DDN} & 0.55 & 0.90 & 0.97 & 0.98 & 0.99 & \bf{1.00} \newline
> > > \text{DMIX} & 0.35 & 0.88 & 0.93 & 0.96 & \bf{1.00} & \bf{1.00} \newline
> > > \end{array}
> > >
> > > ###### $\bf{2s3z-v1}$
> > > \begin{array}{l|cccccc}
> > > \text{Timesteps}  & 2.5 * 10^5 & 5.0 * 10^5 & 7.5 * 10^5 & 10.0 * 10^5 & 15.0 * 10^5 & 20.0 * 10^5 \newline
> > > \hline
> > > \textbf{DRIMA-SS} & 0.54 & \bf{0.92} & 0.97 & \bf{0.98} & \bf{0.98} & 0.99 \newline
> > > \textbf{DRIMA-SN} & 0.45 & \bf{0.92} & 0.92 & 0.92 & 0.91 & 0.95 \newline
> > > \text{VDN} & 0.41 & 0.73 & 0.91 & 0.91 & 0.97 & 0.94 \newline
> > > \text{QMIX} & \bf{0.57} & 0.91 & 0.96 & 0.97 & 0.97 & 0.98 \newline
> > > \text{QPLEX} & 0.46 & \bf{0.92} & \bf{0.98} & \bf{0.98} & 0.97 & \bf{1.00} \newline
> > > \text{DDN} & 0.34 & 0.65 & 0.70 & 0.81 & 0.91 & 0.93 \newline
> > > \text{DMIX} & 0.51 & 0.88 & 0.94 & 0.97 & \bf{0.98} & 0.99 \newline
> > > \end{array}
> > >
> > >
> > >
> > > ###### $\bf{3s5z-v1}$
> > > \begin{array}{l|cccccc}
> > > \text{Timesteps}  & 2.5 * 10^5 & 5.0 * 10^5 & 7.5 * 10^5 & 10.0 * 10^5 & 15.0 * 10^5 & 20.0 * 10^5 \newline
> > > \hline
> > > \textbf{DRIMA-SS} & 0.26 & 0.60 & \bf{0.94} & \bf{0.95} & \bf{0.95} & 0.97 \newline
> > > \textbf{DRIMA-SN} & 0.03 & 0.38 & 0.72 & 0.85 & 0.88 & 0.88 \newline
> > > \text{VDN} & 0.02 & 0.08 & 0.27 & 0.40 & 0.66 & 0.66 \newline
> > > \text{QMIX} & 0.26 & 0.64 & 0.81 & 0.88 & 0.92 & 0.94 \newline
> > > \text{QPLEX} & \bf{0.27} & \bf{0.67} & 0.87 & 0.91 & \bf{0.95} & \bf{0.98} \newline
> > > \text{DDN} & 0.00 & 0.12 & 0.18 & 0.45 & 0.53 & 0.73 \newline
> > > \text{DMIX} & 0.17 & 0.62 & 0.83 & 0.91 & 0.92 & 0.97 \newline
> > > \end{array}
> > >
> > >
> > > ###### $\bf{1c3s5z-v1}$
> > > \begin{array}{l|cccccc}
> > > \text{Timesteps}  & 2.5 * 10^5 & 5.0 * 10^5 & 7.5 * 10^5 & 10.0 * 10^5 & 15.0 * 10^5 & 20.0 * 10^5 \newline
> > > \hline
> > > \textbf{DRIMA-SS} & 0.14 & \bf{0.77} & \bf{0.91} & \bf{0.91} & 0.94 & 0.94 \newline
> > > \textbf{DRIMA-SN} & 0.06 & 0.68 & \bf{0.91} & \bf{0.91} & \bf{0.95} & 0.94 \newline
> > > \text{VDN} & 0.01 & 0.16 & 0.42 & 0.58 & 0.83 & 0.84 \newline
> > > \text{QMIX} & \bf{0.20} & 0.58 & 0.69 & 0.77 & 0.93 & 0.96 \newline
> > > \text{QPLEX} & 0.11 & 0.67 & 0.87 & 0.90 & 0.94 & \bf{0.98} \newline
> > > \text{DDN} &0.04 & 0.20 & 0.40 & 0.53 & 0.60 & 0.71 \newline
> > > \text{DMIX} & 0.13 & 0.70 & 0.84 & \bf{0.91} & 0.90 & 0.95 \newline
> > > \end{array}
> > >
> > >
> > > ###### $\bf{10m\\_vs\\_11m-v1}$
> > > \begin{array}{l|cccccc}
> > > \text{Timesteps}  & 2.5 * 10^5 & 5.0 * 10^5 & 7.5 * 10^5 & 10.0 * 10^5 & 15.0 * 10^5 & 20.0 * 10^5 \newline
> > > \hline
> > > \textbf{DRIMA-SS} & 0.11 & \bf{0.56} & \bf{0.94} & 0.93 & \bf{0.98} & \bf{0.98} \newline
> > > \textbf{DRIMA-SN} & 0.09 & \bf{0.56} & 0.85 & \bf{0.95} & \bf{0.98} & 0.95 \newline
> > > \text{VDN} & 0.01 & 0.15 & 0.46 & 0.59 & 0.91 & 0.95 \newline
> > > \text{QMIX} & 0.07 & 0.22 & 0.76 & 0.87 & 0.94 & 0.95 \newline
> > > \text{QPLEX} & 0.02 & 0.14 & 0.32 & 0.51 & 0.64 & 0.78 \newline
> > > \text{DDN} &0.03 & 0.07 & 0.23 & 0.36 & 0.66 & 0.89 \newline
> > > \text{DMIX} & \bf{0.15} & 0.53 & 0.82 & \bf{0.95} & 0.95 & \bf{0.98} \newline
> > > \end{array}

---

> > > > ### Author Response · Authors · 2021-11-20
> > > > **Response to Reviewer opi3 (4)**
> > > >
> > > >
> > > > ###### $\bf{2c\\_vs\\_64zg-v1}$
> > > > \begin{array}{l|cccccc}
> > > > \text{Timesteps}  & 2.5 * 10^5 & 5.0 * 10^5 & 7.5 * 10^5 & 10.0 * 10^5 & 15.0 * 10^5 & 20.0 * 10^5 \newline
> > > > \hline
> > > > \textbf{DRIMA-SS} & 0.00 & 0.01 & 0.01 & 0.02 & 0.13 & 0.16 \newline
> > > > \textbf{DRIMA-SN} & 0.00 & \bf{0.02} & \bf{0.08} & \bf{0.31} & \bf{0.77} & \bf{0.89} \newline
> > > > \text{VDN} & 0.00 & 0.00 & 0.00 & 0.00 & 0.08 & 0.26 \newline
> > > > \text{QMIX} & 0.00 & 0.00 & 0.05 & 0.09 & 0.23 & 0.42 \newline
> > > > \text{QPLEX} & 0.00 & 0.00 & 0.00 & 0.02 & 0.19 & 0.27 \newline
> > > > \text{DDN} & 0.00 & 0.00 & 0.00 & 0.00 & 0.01 & 0.07 \newline
> > > > \text{DMIX} &0.00 & 0.00 & 0.06 & 0.20 & 0.66 & 0.81 \newline
> > > > \end{array}
> > > >
> > > > ###### $\bf{bane\\_vs\\_bane-v1}$
> > > > \begin{array}{l| cccccc}
> > > > \text{Timesteps}  & 2.5 * 10^5 & 5.0 * 10^5 & 7.5 * 10^5 & 10.0 * 10^5 & 15.0 * 10^5 & 20.0 * 10^5 \newline
> > > > \hline
> > > > \textbf{DRIMA-SS} &  0.90 & \bf{1.00} & \bf{1.00} & \bf{1.00} & \bf{1.00} & \bf{1.00} \newline
> > > > \textbf{DRIMA-SN} &  \bf{0.99} & \bf{1.00} & \bf{1.00} & \bf{1.00} & \bf{1.00} & \bf{1.00} \newline
> > > > \text{DDN} & 0.27 & 0.35 & 0.13 & 0.22 & 0.18 & 0.18 \newline
> > > > \text{DMIX} & 0.52 & 0.79 & 0.93 & 0.95 & 0.99 & 1.00 \newline
> > > > \end{array}
> > > >
> > > > ###### $\bf{3s5z\\_vs\\_3s6z-v1}$
> > > > \begin{array}{l| cccccc}
> > > > \text{Timesteps}  & 2.5 * 10^5 & 5.0 * 10^5 & 7.5 * 10^5 & 10.0 * 10^5 & 15.0 * 10^5 & 20.0 * 10^5 \newline
> > > > \hline
> > > > \textbf{DRIMA-SS} & 0.00 & \bf{0.01} & \bf{0.03} & \bf{0.06} & \bf{0.34} & \bf{0.41} \newline
> > > > \textbf{DRIMA-SN} &0.00 & 0.00 & 0.00 & 0.01 & 0.03 & 0.05 \newline
> > > > \text{VDN} &  0.00 & 0.00 & 0.00 & 0.00 & 0.00 & 0.00 \newline
> > > > \text{QMIX} & 0.00 & 0.00 & 0.00 & 0.00 & 0.00 & 0.00 \newline
> > > > \text{QPLEX} & 0.00 & 0.00 & 0.00 & 0.00 & 0.00 & 0.00 \newline
> > > > \text{DDN} & 0.00 & 0.00 & 0.00 & 0.01 & 0.05 & 0.23 \newline
> > > > \text{DMIX} &  0.00 & 0.00 & 0.02 & 0.00 & 0.03 & 0.17 \newline
> > > > \end{array}
> > > >
> > > > ###### $\bf{6h\\_vs\\_8z-v1}$
> > > > \begin{array}{l|cccccc}
> > > > \text{Timesteps}  & 2.5 * 10^5 & 5.0 * 10^5 & 7.5 * 10^5 & 10.0 * 10^5 & 15.0 * 10^5 & 20.0 * 10^5 \newline
> > > > \hline
> > > > \textbf{DRIMA-SS} & 0.00 & 0.00 & 0.01 & 0.00 & 0.03 & 0.03 \newline
> > > > \textbf{DRIMA-SN} & 0.00 & 0.00 & 0.01 & 0.00 & 0.02 & 0.04 \newline
> > > > \text{VDN} & 0.00 & 0.00 & 0.00 & 0.00 & 0.00 & 0.00 \newline
> > > > \text{QMIX} & 0.00 & 0.00 & 0.01 & 0.02 & 0.02 & 0.02 \newline
> > > > \text{QPLEX} &  0.00 & 0.00 & 0.01 & 0.02 & 0.02 & 0.05 \newline
> > > > \text{DDN} & 0.00 & 0.00 & 0.00 & \bf{0.05} & \bf{0.34} & \bf{0.57} \newline
> > > > \text{DMIX} & 0.00 & \bf{0.01} & \bf{0.02} & 0.00 & 0.02 & 0.07 \newline
> > > > \end{array}
> > > >
> > > > ###### $\bf{27m\\_vs\\_30m-v1}$
> > > > \begin{array}{l|cccccc}
> > > > \text{Timesteps}  & 2.5 * 10^5 & 5.0 * 10^5 & 7.5 * 10^5 & 10.0 * 10^5 & 15.0 * 10^5 & 20.0 * 10^5 \newline
> > > > \hline
> > > > \textbf{DRIMA-SS} & 0.00 & \bf{0.16 }& \bf{0.36} & \bf{0.52} & \bf{0.88} & \bf{0.88} \newline
> > > > \textbf{DRIMA-SN} & 0.00 & 0.11 & 0.31 & 0.50 & 0.78 & 0.85 \newline
> > > > \text{DDN} & 0.00 & 0.01 & 0.02 & 0.04 & 0.09 & 0.30 \newline
> > > > \text{DMIX} & 0.00 & 0.10 & 0.25 & 0.41 & 0.63 & 0.78 \newline
> > > > \end{array}
> > > >
> > > > ###### $\bf{corridor-v1}$
> > > > \begin{array}{l|cccccc}
> > > > \text{Timesteps}  & 2.5 * 10^5 & 5.0 * 10^5 & 7.5 * 10^5 & 10.0 * 10^5 & 15.0 * 10^5 & 20.0 * 10^5 \newline
> > > > \hline
> > > > \textbf{DRIMA-SS} & 0.00 & 0.00 & 0.00 & 0.00 & 0.00 & 0.00 \newline
> > > > \textbf{DRIMA-SN} & 0.00 & 0.00 & 0.00 & 0.00 & 0.00 & 0.00 \newline
> > > > \text{DDN} & 0.00 & 0.00 & 0.00 & 0.00 & 0.00 & \bf{0.03} \newline
> > > > \text{DMIX} & 0.00 & 0.00 & 0.00 & 0.00 & 0.00 & 0.00 \newline
> > > > \end{array}
> > > >
> > > > ---
> > > >
> > > > **Q4. This paper is not well written, and there are various grammatical errors throughout the paper. Some notations are not clear.**
> > > >
> > > > **A4.** Thank you for the helpful suggestion. To alleviate your concern, we thoroughly corrected grammatical errors such as incorrect verb forms, unclear pronoun references, and incorrect usage of articles.
> > > >
> > > > Moreover, to further improve readability, we reorganized the preliminary and methodology section in the revised draft. To be specific, we added the motivation of our work in Section 3.1 and more descriptive explanations in Section 2 to help a broader audience understand our method.

---

### Official Review · Reviewer_xk2m · 2021-11-02

**Correctness:** 3
**Technical Novelty And Significance:** 3
**Empirical Novelty And Significance:** 3
**Recommendation:** 6
**Confidence:** 4

**Main Review:**

Strengths:
- The paper is well-written and appropriately discusses the relevant literature of related MARL algorithms.
- The idea of disentangling agent-wise and environment-wise risks is novel and applying risk-sensitive reinforcement learning in multi-agent reinforcement learning is important to the field.
- The experiments in the difficult StarCraft tasks show an advantage of the proposed method in relation to other state-of-the-art MARL algorithms. The ablation study shows that environment-wise risk-seeking helps in tasks where more exploration is necessary.

Weaknesses:
- I missed a more deep discussion and comparison with DFAC (Sun et al., 2021) and other distributional MARL algorithms. Is the environment-wise risk control in both algorithms the same (that is, following IQN)?  It is not clear whether the only contribution of DRIMA is introducing the agent-wise risk level through the Weighted QMIX loss function.
- The agent-wise risk level w_agt in the loss for non-optimal actions in Section 3.1 seems to only control whether it will follow the loss function from QTRAN (w_agt=1) or from Weighted QMIX. In the latter case, is the w_agr parameter playing the exact same role as the parameter alpha in the Weighted QMIX loss function? If that is the case, the only contribution here is the interpretation of this parameter as a risk level control.
- In the experiments it seems that agent-wise risk-seeking always shows higher performance (that is, following QTRAN loss function with w_agt=1). It is not clear what is the advantage of using different values for w_agt in terms of risk control.
- Although common in the deep MARL literature (especially considering the computationally expensive StarCraft tasks), the paper does not justify why four random seeds are sufficient to make their claims.

Furthermore, I have the following questions and criticisms:

- Section 2.1 is not very self-contained. I needed to read related works (e.g. QMIX) to understand why the decomposition of Eq. 1 can be applied. I suggest the authors improve the discussion on when it is reasonable to expect this condition to be true and its implications.
- In Eq. 3, “y is the true target action-value by unconstrained true action-value estimator Qjt”. Does this mean that y equals Qjt(tau,u), as denoted in Eq. 2? If so, I suggest the authors make it consistent between both equations.
- “Under this paradigm, value-based one has gained impressive attention,”
It is not clear what “one” is referring to in this sentence.
 - “(a) agent-wise and (b) environment-wise, which do not emerge in the single-agent setting”
Environment-wise risk can also occur in the single-agent setting. E.g. a stochastic reward function can sometimes give higher or lower rewards to the agent.
- In the evaluation period, it is possible to choose actions with different risk levels by simply changing the value of w_agt. However, is it possible to control the environment-wise risk level after training? If I understand correctly, this is not possible since the individual agent networks do not depend on w_env.
- “DFAC is another distributional MARL framework with proposed mean-shape decomposition while employing IQN network for agent-wise utility function”
The meaning of mean-shape decomposition is not defined, I believe.
- “Furthermore, (s, u, r, s’) is a tuple of objects collected from consecutive time steps in the Markov decision process”
I believe the authors meant a  tuple of experience transitions. Is that the case? It is not clear what “objects” are, here.


======== UPDATE ========

I have increased my score to "weak accept" after the author's clarifications and novel additions to the draft. They significantly clarified many of my concerns for the theoretical justifications of the agent-wise and environment-wise "risk" parameters.

**Summary Of The Paper:**

This paper proposes a novel multi-agent reinforcement learning algorithm that disentangles randomness/risk sources coming from (i) unobservable actions of cooperative agents and (ii) unobservable actions of enemy agents (environment stochasticity). The proposed method, DRIMA, uses Implicit Quantile Networks (IQN) to learn the joint action action-value distribution and controls the environment-wise risk level by changing its sampling distribution. The agent-wise risk level, on the other hand, is controlled through a hyperparameter in the loss function that softly ignores learning of the action-value function for non-optimal actions, and learns it accurately only for optimal actions in which agents are fully cooperative (similarly to previous approaches, e.g. Weighted QMIX). The proposed method shows improved performance compared to other state-of-the-art MARL algorithms in difficult StarCraft benchmark tasks.

**Summary Of The Review:**

The paper is well-written and the algorithm is well-justified. However, the main technical contributions seem to come from previous approaches (agent-wise risk level from Weighted Q-MIX and environment-wise risk level from DFAC). Although the experiments show an advantage of the proposed method in hard multi-agent tasks, it is critical that the authors better clarify the algorithmic contributions of the proposed method. Especially, it is important to distinguish the risk level controls of DRIMA from the algorithms it was built upon. Hence, my score for the paper in its current form is a “weak reject”. I am willing to raise it if the authors address the concerns pointed in my review.

---

> ### Author Response · Authors · 2021-11-20
> **Response to Reviewer xk2m (1)**
>
> We express our deep appreciation for your time and insightful comments. We address your comments one by one. The revisions made are marked with “$\text{\color{magenta}magenta}$” in the revised paper.
>
> **Q1. I missed a more deep discussion and comparison with DFAC (Sun et al., 2021)**
>
> **A1.** DFAC extends IQN, a single-agent distributional RL algorithm, to learn the randomness of the environment. They propose DDN and DMIX as the DFAC variants of VDN and QMIX, respectively. However, they do not consider the agent-wise risk that arises when learning networks with limited expressive power such as QMIX. From the result in our response 2 below, DMIX has limited capability in adjusting risk-sensitivity. We observe that leveraging risk-sensitivity affects agent-wise and environment-wise risk simultaneously. Therefore, DMIX produces a suboptimal policy in environments requiring different adjustments for each risk source. The respective discussion was in Sections 2 and 3 of our original submission, and we carefully revised it further for readers of interest in the revised draft.
>
> ---
>
> **Q2. Is the $w_{\mathtt{agt}}$ parameter playing the exact same role as the parameter alpha in the Weighted QMIX loss function?**
>
> **A2.** The $w_{\mathtt{agt}}$ in DRIMA and $\alpha$ in the Weighted QMIX have different roles; $w_{\mathtt{agt}}$ controls only agent-wise risk-sensitivity while alpha controls agent-wise and environment-wise risk-sensitivity in an entangled manner. While $w_{\mathtt{agt}}$ adjusts agent-wise risk separately thanks to our proposed hierarchical quantile structure and quantile regression, Weighted QMIX deals with all the risks by using the alpha. We added the related discussion in Section 3.1 of the revised draft.
>
> To support this further, we provided additional experiments on the stochastic two-step matrix game in Section 4 of the revised draft. In this game, two agents select action in {$\{A, B, C\}$} and receive a shared global payoff (reward) as Tables 1 and 2 below show. The payoffs of the first and the second step are designed to assess whether an algorithm is able to handle agent-wise and environment-wise risk, respectively. For the first step, agent-wise risk-seeking (cooperation) is crucial because the optimal action is $(A, A)$ but taking action $A$ is very risky because when the teammate chooses to be non-cooperative (i.e., by selecting action $B$ or $C$), a catastrophic reward of -12 is provided. In the second step, handling environment-wise risks is highlighted since there are various combinations of mean and variance depending on joint actions, i.e., $\mathcal{N}(-1, 10)$ for the action $(C, C)$ and $\mathcal{N}(-1, 0)$ for the action $(B, B)$.
>
> As shown in Table 3 below, DRIMA is able to adjust agent-wise risk-sensitivity via $w_{\mathtt{agt}}$ while the alpha of Weighted QMIX controls agent-wise and environment-wise risk in an entangled manner. Due to their limitations in representing diverse risk-sensitivity, baselines including Weighted QMIX fail to learn the optimal policy; agent-wise risk-seeking and environment-wise risk-neutral show the largest test mean reward. Whereas, DRIMA is capable of learning the optimal policy.

---

> > ### Author Response · Authors · 2021-11-20
> > **Response to Reviewer xk2m (2)**
> >
> > ###### $\textbf{Table1: Payoff in the first step of the two-step stochastic matrix game}$
> > \begin{array}{cc|ccc}
> > & &    & u_{2} & \newline
> > & & A & B & C \newline
> > \hline
> > & A & \mathcal{N}(8, \textcolor{gray}{0}) & \mathcal{N}(-12, \textcolor{gray}{0}) & \mathcal{N}(-12, \textcolor{gray}{0}) \newline
> > u_{1} & B & \mathcal{N}(-12, \textcolor{gray}{0})  & \mathcal{N}(0, \textcolor{gray}{0}) & \mathcal{N}(0, \textcolor{gray}{0}) \newline
> > & C & \mathcal{N}(-12, \textcolor{gray}{0})  & \mathcal{N}(0, \textcolor{gray}{0}) & \mathcal{N}(0, \textcolor{gray}{0}) \newline
> > \end{array}
> >
> > ###### $\textbf{Table2: Payoff in the second step of the two-step stochastic matrix game}$
> > \begin{array}{cc|ccc}
> > & &    & u_{2} & \newline
> > & & A & B & C \newline
> > \hline
> > & A & \mathcal{N}(1, 5) & \mathcal{N}(0, 5) & \mathcal{N}(0, 5) \newline
> > u_{1} & B & \mathcal{N}(0, 5)  & \mathcal{N}(-1, 0) & \mathcal{N}(-1, 5) \newline
> > & C & \mathcal{N}(0, 5)  & \mathcal{N}(-1, 5) & \mathcal{N}(-1, 10) \newline
> > \end{array}
> >
> > ###### $\textbf{Table3: Test rewards and trained policy in the stochastic two-step matrix game}$
> > \begin{array}{l|ccc cc cc}
> > \text{Algorithm} & & \text{Test reward} & & & & & & \newline
> > \text{(agent-wise risk, env-wise risk)} & \text{Min} & \text{Mean} & \text{Max} & 1^{\text{st}} ~ u_{1}& 1^{\text{st}} ~ u_{2} & 2^{\text{nd}} ~ u_{1} & 2^{\text{nd}} ~ u_{2}\newline
> > \hline
> > \text{DRIMA (Neutral, Averse)} & -1.00 & -1.00 & -1.00 & \text{B or C} & \text{B or C} & \text{B} & \text{B} & \newline
> > \text{DRIMA (Neutral, Neutral)} & -1.28 & 0.89 & 4.13 & \text{B or C} & \text{B or C} & \text{A} & \text{A} \newline
> > \text{DRIMA (Neutral, Seeking)} & -6.65 & 2.34 & 10.99 & \text{B or C} & \text{B or C} & \text{C} & \text{C} \newline
> > \text{DRIMA (Seeking, Averse)} & \textbf{7.00} & 7.00 & 7.00 & \text{A} & \text{A} & \text{B} & \text{B} \newline
> > \text{DRIMA (Seeking, Neutral)} & 6.03 & \textbf{8.98} & 11.07 & \text{A} & \text{A} & \text{A} & \text{A} \newline
> > \text{DRIMA (Seeking, Seeking)} & 2.78 & 5.50 & \textbf{13.01} & \text{A} & \text{A} & \text{C} & \text{C} \newline
> > \hline
> > \text{DMIX (Averse)} & -1.00 & -1.00 & -1.00 & \text{B or C} & \text{B or C} & \text{B} & \text{B} \newline
> > \text{DMIX (Neutral)} & -2.66 & 0.89 & 2.94 & \text{B or C} & \text{B or C} & \text{A} & \text{A}  \newline
> > \text{DMIX (Seeking)} & 1.89 & 6.96 & \textbf{12.81} & \text{A} & \text{A} & \text{C} & \text{C} \newline
> > \hline
> > \text{OW-QMIX (Neutral)} & -3.04 & 0.63 & 2.98 & \text{B or C} & \text{B or C} & \text{A} & \text{A} \newline
> > \text{OW-QMIX (Seeking)} & 1.95 & 6.94 & \textbf{12.87} & \text{A} & \text{A} & \text{C} & \text{C}  \newline
> > \hline
> > \text{QMIX (-)}& -0.65 & 0.61 & 1.97 & \text{B or C} & \text{B or C} & \text{A} & \text{A} \newline
> > \text{QTRAN (-)} & 5.92 & 9.12 & 11.31 & \text{A} & \text{A} & \text{A} & \text{A} \newline
> > \text{QPLEX (-)} & 6.17 & 8.96 & 10.54 & \text{A} & \text{A} & \text{A} & \text{A} \newline
> > \end{array}
> >
> > ---
> >
> > **Q3. In the experiments, it seems that agent-wise risk-seeking always shows higher performance**
> >
> > **A3.** As you said, in the SMAC environment, agent-wise risk-seeking always performs well because assuming “cooperative” teammates in training helps to learn tactics to beat enemies. However, in the real-world, adversarial teammates or randomly behaving agents are likely to exist, and we expect that agent-wise risk-neutral policy can perform better in such situations. This highlights the importance of disentangling risk-sources by DRIMA. To demonstrate the importance of an agent-wise risk-neutral policy, we increased the randomness of the agents’ behavior in additional experiments with simple one-step matrix game as shown in Table 4 below. If the agents act randomly with a 50% probability even during the test phase, the agent-wise risk-neutral policy shows higher performance than the agent-wise risk-seeking policy as shown in Table 5 below. We added related discussions in Appendix E in the revised draft.
> >
> > ###### $\textbf{Table4: Payoff matrix of the one-step game}$
> > \begin{array}{cc|ccc}
> > & &    & u_{2} & \newline
> > & & A & B & C \newline
> > \hline
> > & A & 8 & -100 & -100 \newline
> > u_{1} & B & -100 & 0 & 0 \newline
> > & C & -100 & 0 & 0 \newline
> > \end{array}
> >
> > ###### $\textbf{Table5: Test rewards with noisy agents in the one-step matrix game}$
> > \begin{array}{l|c}
> > \text{Algorithm} & \text{Test reward} \newline
> > \text{(agent-wise risk, env-wise risk)} & \text{Mean} \newline
> > \hline
> > \text{DRIMA (Neutral, Neutral)} & \textbf{-26.92} \newline
> > \text{DRIMA (Seeking, Neutral)} & -45.01 \newline
> > \hline
> > \text{DMIX (Neutral)} & -27.84 \newline
> > \text{DMIX (Seeking)} & -45.64 \newline
> > \hline
> > \text{OW-QMIX (Neutral)} & -27.52 \newline
> > \text{OW-QMIX (Seeking)} & -44.48 \newline
> > \end{array}

---

> > > ### Author Response · Authors · 2021-11-20
> > > **Response to Reviewer xk2m (3)**
> > >
> > > **Q4. Although common in the deep MARL literature, the paper does not justify why four random seeds are sufficient to make their claims**
> > >
> > > **A4.** Thank you for the helpful suggestion. Our experimental results show not only the median value but also the confidence interval. This confidence interval demonstrates that DRIMA outperforms other baselines in a reliable error range.  Nevertheless, for the next revised version of our paper, we would like to try our best to increase our random seeds to five, following the prior work (DFAC).
> > >
> > > ---
> > >
> > > **Q5. Section 2.1 is not very self-contained (e.g. QMIX).**
> > >
> > > **A5.** Thank you for the suggestion. To address your concern, we added a description of VDN and QMIX in Section 2.1 in the revised draft. For example, we added explanations about the structural constraints of VDN and how QMIX extends VDN.
> > >
> > > ---
> > >
> > > **Q6. I suggest the authors make notation consistent between Equation 2 and 3.**
> > >
> > > **A6.** We clarify that y is a notation mainly used to express a fixed target in the reinforcement learning community, and it is defined in the form of $y = r + \gamma Q_\mathtt{jt}^\mathtt{target} (s', \bf{\tau}', {{\bf{u'}}_{\mathtt{opt}}})$ in Weighted QMIX. In the revised draft, we further clarified this in Section 2.1
> > >
> > > ---
> > >
> > > **Q7. It is not clear what “one” is referring to in this sentence “Under this paradigm, value-based one has gained impressive attention.**
> > >
> > > **A7.** Thank you for the helpful suggestion. To alleviate your concern, we state the ‘value-based CTDE method’ in our revised draft.
> > >
> > > ---
> > >
> > > **Q8. Environment-wise risk can also occur in the single-agent setting.**
> > >
> > > **A8.** Thank you for the suggestion. Our writing was unclear, which leads to such misunderstanding. To clarify this, we now stated that there exists only environment-wise risk in the single-agent setting in the revised draft.
> > >
> > > ---
> > >
> > > **Q9. The meaning of mean-shape decomposition is not defined**
> > >
> > > **A9.** Thank you for the suggestion. To address your concern, we added a description of DFAC in Section 2.2 in the revised draft. For example, we added a description that DFAC separates mean and variance part of utility function to handle the value function factorization through mean-shape decomposition
> > >
> > > ---
> > >
> > > **Q10. It is not clear what “objects” are in Section 2.2**
> > >
> > > **A10.** We agree that the word “object” is not clear. Following your suggestion, we modified the word “objects” by “experience transitions”.

---

> > > > ### Comment · Reviewer_xk2m · 2021-11-21
> > > > **Response to Authors**
> > > >
> > > > I thank the authors for the detailed response and for carefully taking into consideration the reviewer's suggestions in the revised draft. The two-step matrix game indeed improved the clarity of the contributions.
> > > >
> > > > However, my main concern is still on how the parameters $w_{agt}$ and $w_{env}$ are used and the theoretical justification for claiming they are controlling risk level controllers. Specifically:
> > > >
> > > > * Are $w_{agt}$ and $w_{env}$ used only when training the networks? Is it possible to control the risk levels in the evaluation step?
> > > > The authors claim "the feed forward network is not conditioned on $w_{agt}$". It is not clear which/whether networks are parameterized by $w_{agt}$ and $w_{env}$ or if they control only the loss functions.
> > > >
> > > > * In the Weigthed QMIX paper, $\alpha$ has a completly different theoretical justification than $w_{agt}$. Precisely, it corrects the argmax of the joint Q function to satisfy the monotonic conditions. I do not think it is possible to claim that $w_{agt}$ controls whether agents are taking risks or not. Can you elaborate and clarify these claims?

---

> > > > > ### Author Response · Authors · 2021-11-23
> > > > > **Response to Reviewer xk2m (4)**
> > > > >
> > > > > We express our deep appreciation for your time and insightful comments. The revisions made are marked with “$\text{\color{magenta} magenta}$” in the revised paper.
> > > > >
> > > > > **Q1-1. Are $w_\mathtt{agt}$ and $w_\mathtt{env}$ used only when training the networks? Is it possible to control the risk levels in the evaluations step?**
> > > > >
> > > > > **A1-1.** Both $w_\mathtt{agt}$ and $w_\mathtt{env}$ are only used when training the networks. Therefore, it is impossible to control the risk level in the evaluation step.
> > > > >
> > > > > Specifically, the agent-wise risk-level cannot be controlled in the evaluation step since true action-value and optimal policy are estimated with predetermined risk-levels. We note that this is not just a problem with our method, but also exists in the single-agent distributional reinforcement learning algorithms such as IQN. We think it is an interesting research direction to make risk-level adjustable in the evaluation step.
> > > > >
> > > > > **Q1-2. The authors claim “the feed-forward network is not conditioned on $w_\mathtt{agt}$”. It is not clear which/whether networks are parameterized by $w_\mathtt{agt}$ and $w_\mathtt{env}$ or if they control only the loss functions.**
> > > > >
> > > > > **A1-2.** The feed-forward network (the true action-value estimator) is conditioned on $w_\mathtt{env}$, not on $w_\mathtt{agt}$ while the transformed action-value estimator is conditioned on $w_\mathtt{agt}$, not on $w_\mathtt{env}$. In addition, $w_\mathtt{agt}$ control loss functions as a weighting term for training the transformed action-value estimator. In the revised draft, we improve Figure 3 (Architecture of DRIMA) to enhance clarity.

---

> > > > > > ### Author Response · Authors · 2021-11-23
> > > > > > **Response to Reviewer xk2m (5)**
> > > > > >
> > > > > > **Q2-1. In the Weighted QMIX paper, $\alpha$ has a completely different theoretical justification than $w_\mathtt{agt}$. Precisely, it corrects the argmax of the joint Q function to satisfy the monotonic conditions.**
> > > > > >
> > > > > > **A2-1.** Although $\alpha$ in the Weighted QMIX comes from a different motivation in the original paper as you mentioned, we find that the $\alpha$ in the Weighted QMIX and $w_\mathtt{agt}$ in DRIMA play the same role: adjusting agent-wise risk-sensitivity. Therefore, $w_\mathtt{agt}$ in DRIMA can control agent-wise risk as the Weighted QMIX did.
> > > > > >
> > > > > > The reason why $\alpha$ in the Weighted QMIX takes such a role can be found from the tight connection between the Weighted QMIX and hysteretic Q-learning [1, 2, 3], which is a representative optimistic (agent-wise risk-seeking) algorithm in a fully distributed MARL. To be specific, one can find the connection in their loss functions. Both of them apply weights to the losses according to the sign of the TD-error for each sample. The weighting mechanism according to the sign of the TD-error assigns a high weight for optimistic (agent-wise risk-seeking) returns.
> > > > > >
> > > > > > Next, to investigate similarities and differences between DRIMA and OW-QMIX to handle risk-sensitivity, we employ a one-step single-state matrix game. In this game, there is only one state, and all episodes finish after a single time-step. We find a connection between the DRIMA and OW-QMIX via the following proposition:
> > > > > >
> > > > > > *Consider a deterministic-reward environment with a single state that terminated immediately. Then the loss ${L_{\mathtt{nopt}}}$ of DRIMA with parameters $w_\mathtt{agt}$ is equivalent to the Weighted QMIX with $\alpha = 2 * (1 - w_\mathtt{agt})$.*
> > > > > >
> > > > > > We prove this proposition in Appendix C.3 in our revised draft. This proposition says that DRIMA and OW-QMIX become equivalent in a deterministic-reward environment.
> > > > > >
> > > > > > **Q2-2. I do not think it is possible to claim that $w_\mathtt{agt}$ controls whether agents are taking risks or not.**
> > > > > >
> > > > > > **A2.** Based on underlying similarity between DRIMA and OW-QMIX (as discussed in Response 2-1), it is possible for $w_\mathtt{agt}$ to control agent-wise risk-sensitivity as OW-QMIX did. In addition, DRIMA can disentangle risk-sensitivity while OW-QMIX cannot.
> > > > > >
> > > > > > To raise further understanding about the ability to disentangle risks by DRIMA and OW-QMIX, we provide theoretical analysis on stochastic environments. In such environments, optimistic algorithms can induce misplaced optimism towards uncontrollable environment dynamics, leading to suboptimal behavior. The optimistic MARL approaches ignore low returns, which are assumed to be caused by teammates' exploratory actions. This causes severe overestimation of action-values in stochastic domains. Since reward and next state in the TD-target $y$ in Weighted QMIX include randomness of the environment, agent-wise risk cannot be extracted separately enough from the sign of the TD-error. Therefore, in DRIMA, we do not use the target $y$ as done in Weighted QMIX. Instead, we use $\mathbb{E_{w_{\mathtt{env}}}}[Z_\mathtt{jt}(w_{\mathtt{env}})]$ to exclude the randomness of the environment in the target. The true action-value estimator $Z_\mathtt{jt}$ of DRIMA does not have any structural restrictions such as monotonicity in value factorization methods. Therefore, $Z_\mathtt{jt}$ can accurately represent only the randomness of the environment. Section 4 and Appendix E.2 supports our claim.
> > > > > >
> > > > > >
> > > > > > **Reference**
> > > > > >
> > > > > > [1] Rashid, Tabish, et al. "Weighted QMIX: Expanding Monotonic Value Function Factorisation for Deep Multi-Agent Reinforcement Learning." NeurIPS. 2020.
> > > > > >
> > > > > > [2] Matignon, Laëtitia, Guillaume J. Laurent, and Nadine Le Fort-Piat. "Hysteretic q-learning: an algorithm for decentralized reinforcement learning in cooperative multi-agent teams." 2007 IEEE/RSJ International Conference on Intelligent Robots and Systems. IEEE, 2007.
> > > > > >
> > > > > > [3] Shayegan Omidshafiei, Jason Pazis, Christopher Amato, Jonathan P. How, and John Vian.  Deep decentralized multi-task multi-agent reinforcement learning under partial observability. ICML 2017.

---

> > > > > > > ### Comment · Reviewer_xk2m · 2021-11-23
> > > > > > > **Response to Authors**
> > > > > > >
> > > > > > > I appreciate the author's response and novel additions/modifications to the text. Henceforth, I am increasing my score to "weak accept".
> > > > > > > I am unable to further increase my score due to the clarity and organization of the draft. For instance, a good understanding of the parameter $w_{agt}$ as an agent-wise risk-level controller (which is a critical part of the proposed method) is only provided in the newly added Appendix C.3.
> > > > > > > The formal definitions of risk (as replied to the reviewer 5LTK) are also fundamental and should appear in the paper.
> > > > > > >
> > > > > > > Ps: Because the algorithm has many implementation subtleties, I believe providing the source code would be necessary for proper reproducibility.

---

> > > > > > > > ### Author Response · Authors · 2021-11-30
> > > > > > > > **We highly appreciate your insightful and helpful comments.**
> > > > > > > >
> > > > > > > > We highly appreciate your insightful and helpful comments. We are glad to hear that our response helped you understand our method. Thank you very much for raising the score.
> > > > > > > >
> > > > > > > > p.s. In order to promote reproducibility, our code is already submitted in the supplementary material. We will also release our source codes on Github after the review period.

---

### Official Review · Reviewer_5LTK · 2021-11-04

**Correctness:** 4
**Technical Novelty And Significance:** 3
**Empirical Novelty And Significance:** 2
**Recommendation:** 6
**Confidence:** 3

**Main Review:**

STRENGTHS

The motivation behind DRIMA is intuitive and interesting, and the implementation is non-trivial. The authors compare to a wide range of appropriate baselines on a challenging community benchmark, and achieve strong and convincing performance.

WEAKNESSES

The experiments in the paper jump straight to the SMAC domain, without testing their approach in a diagnostic illustrative environment. This made it difficult to understand some aspects of their model. For example, the authors find in Fig 6 that DRIMA seems to perform best when risk-seeking in terms of both agent-wise and environment-wise risk. This runs counter to the authors stated motivation of designing agents that can be optimistic wrt teammates but more risk-neutral wrt the environment. Unfortunately, the authors don't seem to find other tasks/environments for which different risk balances are optimal. This raises the question - is it always better for DRIMA agents to be risk-seeking in both regards? If so, why is it important to learn a factorizied representation of uncertainty anyway? A diagnostic illustrative environment would allow the authors to explore these questions, and would result in more clear scientific contribution of the paper.

Another example question of mine that went unanswered - presumably the reason to be risk-seeking wrt teammates and risk-neutral wrt the environment is that teammates are learning and so can be expected to prosocially adapt. For this reason, would DRIMA perform better than baselines if one teammate learnt slower than the other, or if their learning were reset part way through training? The intuition is that DRIMA might be better at helping lagging teammates "catch up" due to this optimism.

Also, although reviewers are instructed not to penalize papers for writing style, the current draft suffers from enough grammatical errors so as to seriously damage readability. Thus the paper would benefit from significant editorial review.

Lastly, the intuition that uncertainty and optimism over other agents can help facilitate cooperation was also discussed in Baker 2020 (https://arxiv.org/abs/2011.05373), which the authors might find interesting.

**Summary Of The Paper:**

The authors introduce DRIMA - a distributional CTDE multi-agent RL approach that separately learns to model return stochasticity arising from other agents vs the environment. They argue this is useful because for example it allows agent to be optimistic wrt teammates (who can prosocially adapt) but more risk-neutral wrt the environment (which does not adapt). They evaluate the effectiveness of their approach on the Starcraft MultiAgent Challenge (SMAC), and outperform several state-of-the-art baselines.

**Summary Of The Review:**

As is, I found the results interesting and convincing enough to be just over the bar for acceptance. However, I am fairly uncertain in my assessment in part due to a lack of familiarity with the relevant distributional RL literature. In any case, I would feel more strongly about acceptance if the writing clarity were improved and additional diagnostic explanatory experiments were added, as outlined above.

---

> ### Author Response · Authors · 2021-11-20
> **Response to Reviewer 5LTK (1)**
>
> We express our deep appreciation for your time and insightful comments. We address your comments one by one. The revisions made are marked with “$\text{\color{magenta}magenta}$” in the revised paper.
>
> **Q1. A diagnostic illustrative environment would result in a more clear scientific contribution of the paper.**
>
> **A1.** Thank you for the suggestion. To incorporate your comment, we evaluate DRIMA and baselines in a simple stochastic two-step matrix game. In this game, two agents select an action in {$\{A, B, C\}$} and receive a shared global payoff (reward) as Tables 1 and 2 show. The payoffs of the first and the second steps are designed to assess whether an algorithm is able to handle agent-wise and environment-wise risks, respectively. For the first step, agent-wise risk-seeking (cooperation) is crucial because the optimal action is $(A, A)$ but taking action $A$ is very risky because when the teammate chooses to be non-cooperative (i.e., by selecting action $B$ or $C$), a catastrophic reward of -12 is provided. In the second step, handling environment-wise risks is highlighted since there are various combinations of mean and variance depending on joint actions, i.e., $\mathcal{N}(-1, 10)$ for the action $(C, C)$ and $\mathcal{N}(-1, 0)$ for the action $(B, B)$.
>
> As shown in Table 3, we observe that DRIMA is able to adjust agent-wise and environment-wise risk separately, so it can achieve the optimal policy that requires agent-wise risk-seeking and environment-wise risk-neutral policy (policy whose mean test reward is maximum). Note that other baselines including DMIX, Weighted QMIX, QMIX, QTRAN, and QPLEX have a limitation in adjusting risk-sensitivity, so they fail to represent the optimal policy.
>
> We provide more environmental details and analysis in Section 4 in our revised draft.
>
> ###### $\textbf{Table1: Payoff in the first step of the two-step stochastic matrix game}$
> \begin{array}{cc|ccc}
> & &    & u_{2} & \newline
> & & A & B & C \newline
> \hline
> & A & \mathcal{N}(8, \textcolor{gray}{0}) & \mathcal{N}(-12, \textcolor{gray}{0}) & \mathcal{N}(-12, \textcolor{gray}{0}) \newline
> u_{1} & B & \mathcal{N}(-12, \textcolor{gray}{0})  & \mathcal{N}(0, \textcolor{gray}{0}) & \mathcal{N}(0, \textcolor{gray}{0}) \newline
> & C & \mathcal{N}(-12, \textcolor{gray}{0})  & \mathcal{N}(0, \textcolor{gray}{0}) & \mathcal{N}(0, \textcolor{gray}{0}) \newline
> \end{array}
>
> ###### $\textbf{Table2: Payoff in the second step of the two-step stochastic matrix game}$
> \begin{array}{cc|ccc}
> & &    & u_{2} & \newline
> & & A & B & C \newline
> \hline
> & A & \mathcal{N}(1, 5) & \mathcal{N}(0, 5) & \mathcal{N}(0, 5) \newline
> u_{1} & B & \mathcal{N}(0, 5)  & \mathcal{N}(-1, 0) & \mathcal{N}(-1, 5) \newline
> & C & \mathcal{N}(0, 5)  & \mathcal{N}(-1, 5) & \mathcal{N}(-1, 10) \newline
> \end{array}
>
> ###### $\textbf{Table3: Test rewards and trained policy in the stochastic two-step matrix game}$
> \begin{array}{l|ccc cc cc}
> \text{Algorithm} & & \text{Test reward} & & & & & & \newline
> \text{(agent-wise risk, env-wise risk)} & \text{Min} & \text{Mean} & \text{Max} & 1^{\text{st}} ~ u_{1}& 1^{\text{st}} ~ u_{2} & 2^{\text{nd}} ~ u_{1} & 2^{\text{nd}} ~ u_{2}\newline
> \hline
> \text{DRIMA (Neutral, Averse)} & -1.00 & -1.00 & -1.00 & \text{B or C} & \text{B or C} & \text{B} & \text{B} & \newline
> \text{DRIMA (Neutral, Neutral)} & -1.28 & 0.89 & 4.13 & \text{B or C} & \text{B or C} & \text{A} & \text{A} \newline
> \text{DRIMA (Neutral, Seeking)} & -6.65 & 2.34 & 10.99 & \text{B or C} & \text{B or C} & \text{C} & \text{C} \newline
> \text{DRIMA (Seeking, Averse)} & \textbf{7.00} & 7.00 & 7.00 & \text{A} & \text{A} & \text{B} & \text{B} \newline
> \text{DRIMA (Seeking, Neutral)} & 6.03 & \textbf{8.98} & 11.07 & \text{A} & \text{A} & \text{A} & \text{A} \newline
> \text{DRIMA (Seeking, Seeking)} & 2.78 & 5.50 & \textbf{13.01} & \text{A} & \text{A} & \text{C} & \text{C} \newline
> \hline
> \text{DMIX (Averse)} & -1.00 & -1.00 & -1.00 & \text{B or C} & \text{B or C} & \text{B} & \text{B} \newline
> \text{DMIX (Neutral)} & -2.66 & 0.89 & 2.94 & \text{B or C} & \text{B or C} & \text{A} & \text{A}  \newline
> \text{DMIX (Seeking)} & 1.89 & 6.96 & \textbf{12.81} & \text{A} & \text{A} & \text{C} & \text{C} \newline
> \hline
> \text{OW-QMIX (Neutral)} & -3.04 & 0.63 & 2.98 & \text{B or C} & \text{B or C} & \text{A} & \text{A} \newline
> \text{OW-QMIX (Seeking)} & 1.95 & 6.94 & \textbf{12.87} & \text{A} & \text{A} & \text{C} & \text{C}  \newline
> \hline
> \text{QMIX (-)}& -0.65 & 0.61 & 1.97 & \text{B or C} & \text{B or C} & \text{A} & \text{A} \newline
> \text{QTRAN (-)} & 5.92 & 9.12 & 11.31 & \text{A} & \text{A} & \text{A} & \text{A} \newline
> \text{QPLEX (-)} & 6.17 & 8.96 & 10.54 & \text{A} & \text{A} & \text{A} & \text{A} \newline
> \end{array}

---

> > ### Author Response · Authors · 2021-11-20
> > **Response to Reviewer 5LTK (2)**
> >
> > **Q2. Is it always better for DRIMA agents to be risk-seeking in both regards?**
> >
> > **A2.** As you said, in SMAC environments, agent-wise risk-seeking always performs well because assuming “cooperative” teammates in training helps to learn tactics to beat enemies. However, in the real-world, adversarial teammates or randomly behaving agents are likely to exist, and we expect that agent-wise risk-neutral policy can perform better in such situations. This highlights the importance of disentangling risk sources by DRIMA. To demonstrate the importance of an agent-wise risk-neutral policy, we increased the randomness of the agents’ behavior in additional experiments with a simple one-step matrix game as shown in Table 4 below. If the agents act randomly with a 50% probability even during the test phase, the agent-wise risk-neutral policy shows higher performance than the agent-wise risk-seeking policy as shown in Table 5 below. We added related discussion in Appendix E in our revised draft.
> >
> > ###### $\textbf{Table4: Payoff matrix of the one-step game}$
> > \begin{array}{cc|ccc}
> > & &    & u_{2} & \newline
> > & & A & B & C \newline
> > \hline
> > & A & 8 & -100 & -100 \newline
> > u_{1} & B & -100 & 0 & 0 \newline
> > & C & -100 & 0 & 0 \newline
> > \end{array}
> >
> > ###### $\textbf{Table5: Test rewards with noisy agents in the one-step matrix game}$
> > \begin{array}{l|c}
> > \text{Algorithm} & \text{Test reward} \newline
> > \text{(agent-wise risk, env-wise risk)} & \text{Mean} \newline
> > \hline
> > \text{DRIMA (Neutral, Neutral)} & \textbf{-26.92} \newline
> > \text{DRIMA (Seeking, Neutral)} & -45.01 \newline
> > \hline
> > \text{DMIX (Neutral)} & -27.84 \newline
> > \text{DMIX (Seeking)} & -45.64 \newline
> > \hline
> > \text{OW-QMIX (Neutral)} & -27.52 \newline
> > \text{OW-QMIX (Seeking)} & -44.48 \newline
> > \end{array}
> >
> >
> > ---
> >
> > **Q3. For this reason, would DRIMA perform better than baselines if one teammate learned slower than the other, or if their learning were reset part way through training?**
> >
> > **A3.** In the SMAC environment, we did not observe the slow learning of a specific agent because the agents share parameters. Instead, in the aforementioned stochastic two-step matrix game, one can understand that DRIMA performs better than baseline by helping lagging teammates “catch up” via optimism. To be specific, in the matrix game, lagging teammates are caused by using a full exploration scheme (i.e., $\epsilon=1$ in $\epsilon$-greedy for guaranteeing all available states to be visited), where it can be seen that other agents learn very slowly from the view of one agent.
> >
> > ---
> >
> > **Q4. The current draft suffers from enough grammatical errors so as to seriously damage readability.**
> >
> > **A4.** Thank you for the helpful suggestion. To alleviate your concern, we thoroughly corrected grammatical errors such as incorrect verb forms, unclear pronoun references, and incorrect usage of articles.
> >
> > Moreover, to further improve readability, we reorganized the preliminary and methodology sections in the revised draft. To be specific, we added the motivation of our work in Section 3.1 and more descriptive explanations in Section 2 to help a broader audience understand our method.
> >
> > ---
> >
> > **Q5. Lastly, the intuition that uncertainty and optimism over other agents can help facilitate cooperation was also discussed in Baker 2020 (https://arxiv.org/abs/2011.05373)**
> >
> > **A5.** Thanks for the useful suggestion. The paper you mentioned proposes a method in a fully decentralized setting where rewards are not shared, but learning cooperation through optimism has something in common with ours. We understand that they have very interesting connections. We added the related discussion in Appendix C.2 in our revised draft. Thank you!

---

> > > ### Comment · Reviewer_5LTK · 2021-11-21
> > > **Thank you for the updates**
> > >
> > > Thank you for your thorough response, and the updated results. I think the matrix game results especially are a step in the right direction for interpretability.
> > >
> > > I do however share the concern that other reviewers raised about the paper not being entirely clear about how the differential risk representation is achieved, and agree with them that the paper would benefit from a more formal treatment there and/or an illustrative experiment that does a bit more policy analysis / visualization to help understand this.
> > >
> > > I am keeping my score the same for now (weak accept) but will watch how the discussion with other reviewers plays out and remain open to raising my score.

---

> > > > ### Author Response · Authors · 2021-11-23
> > > > **Response to Reviewer 5LTK (3)**
> > > >
> > > > We express our deep appreciation for your time and insightful comments. The revisions made are marked with “$\text{\color{magenta} magenta}$” in the revised paper.
> > > >
> > > > **Q1. I do however share the concern that other reviewers raised about the paper not being entirely clear about how the differential risk representation is achieved, agree with them that the paper would benefit from a more formal treatment there**
> > > >
> > > > **A1.** To describe risks more formally, one can formulate random variable $Z_{\mathtt{env}}^{\pi}(s, u)$ and $Z_{\mathtt{agt}}(s, u)$ that model environment-wise randomness and agent-wise randomness, respectively. The environment-wise random variable $Z_{\mathtt{env}}^{\pi}(s, u)$ is the sum of future discounted reward given “deterministic” agent-wise policy $\pi$ as follows:
> > > >
> > > > $Z_{\mathtt{env}}^{\pi}(s, u) = \Sigma_{t \geq 0} \gamma^t r(s_t,u_t)|_\pi$,
> > > >
> > > > where $\pi$ is a given deterministic policy. The random variable $Z_{\mathtt{env}}$ depends on immediate stochastic reward and stochastic state transition dynamics, which are “environmental”. Note that agent-wise randomness does not exist in the $Z_{\mathtt{env}}$ due to the given deterministic agent-wise policy.
> > > >
> > > > Meanwhile, the agent-wise random variable, which exists in MARL, implies how the other agents behave for myself to achieve high global reward as follows.
> > > >
> > > > $Z_{\mathtt{agt}}(s, u) = \arg \min_{Z’}  \mathbb{E_{\tau \sim \pi}} \[w(s,u) * (Z'(s, u) - Z_{\mathtt{env}}^{\pi}(s, u))^2\]$,
> > > >
> > > > where $\tau$ is a sampled trajectory and $w(s,u)$ is a weight function depending on agent-wise risk-level. The weight function adjusts agent-wise risk-sensitivity by selecting samples whose agent-wise random variable $Z_{\mathtt{agt}}(s, u)$ are close to the environment-wise random variable $Z_{\mathtt{env}}^{\pi}(s, u)$.
> > > > For example, hysteretic Q-learning, Weighted QMIX, and DRIMA assign a higher weighting to joint actions that could be the true optimal actions for agent-wise optimism.

---

### Author Response · Authors · 2021-11-20
**General Response**

Dear reviewers,

First of all, we express our deepest gratitude for your constructive feedback and valuable comments.

In response to the questions and concerns you raised, we have carefully revised and enhanced our paper with the following additional experiments, writings, and discussions:

- Experiments with a diagnostic illustrative environment that demonstrate the advantages of disentangling two risk sources for all reviewers (Section 4, and Appendix E)
- Reorganization and correction of writing to improve readability for all reviewers (Mostly in Sections 2 and 3)
- Clearer explanations of DRIMA compared to baselines for reviewers xk2m, opi3, and LvSz (Sections 2, 3, and Appendix C)
- More explanations about agent-wise utility for reviewer DnPu (Section 3)
- Experiments with a diagnostic illustrative environment that show the higher performance of agent-wise risk-neutral agents for reviewers 5LTK and xk2m (Appendix E)
- Experiments on more diverse environments, i.e., 14 Maps of SMAC benchmark for reviewer opi3 (Appendix F)
- Corrections of unclear or duplicate notations for reviewer xk2m and opi3 (Section 2)
- Discussion of other related papers on the relationship between uncertainty and optimism for reviewers 5LTK and LvSz (Sections 2, 3, and Appendix C)

The revisions made are marked with “$\text{\color{magenta}magenta}$” in the revised paper.

Thanks, Authors.

---

### Comment · Area_Chair_5jgt · 2021-11-20
**Please read responses from the reviewers**

Dear reviewers,

Please read the detailed responses from the authors. How do they change your evaluation? Do you still have major concerns? Thank you.

---

### Decision · Program_Chairs · 2022-01-20

**Decision:**

Reject

**Comment:**

This paper proposes a method of multi-agent reinforcement learning that separately deals with the risk associated with uncertainties of the other agents and the risk associated with the uncertainties of the environment. This allows for example to be agent-wise risk seeking and environment-wise risk averse.  The proposed approach is largely heuristic with little theoretical justifications.  The experimental results are promising but not sufficiently convincing, given the lack of formalism.  Further improvement on clarity might complement the lack of formalism or theoretical justifications.